# OmicVerse: a framework for bridging and deepening insights across bulk and single-cell sequencing

Zehua Zeng [1,2,9] ✉, Yuqing Ma[3,4,9], Lei Hu [1,5,9], Bowen Tan [6,7], Peng Liu[1], Yixuan Wang[1], Cencan Xing [1,2] ✉, Yuanyan Xiong [8] ✉ & Hongwu Du [1,2] ✉

Single-cell sequencing is frequently affected by "omission" due to limitations in sequencing throughput, yet bulk RNA-seq may contain these ostensibly "omitted" cells. Here, we introduce the single cell trajectory blending from Bulk RNA-seq (BulkTrajBlend) algorithm, a component of the OmicVerse suite that leverages a Beta-Variational AutoEncoder for data deconvolution and graph neural networks for the discovery of overlapping communities. This approach effectively interpolates and restores the continuity of "omitted" cells within single-cell RNA sequencing datasets. Furthermore, OmicVerse provides an extensive toolkit for both bulk and single cell RNA-seq analysis, offering seamless access to diverse methodologies, streamlining computational processes, fostering exquisite data visualization, and facilitating the extraction of significant biological insights to advance scientific research.

Single-cell RNA sequencing (scRNA-seq) and bulk RNA sequencing (RNA-seq) have emerged as essential techniques for exploring cellular heterogeneity, differentiation, and disease mechanisms[1–6]. These technologies facilitate numerous applications, including converting bulk-seq data into single-seq analyses[7], performing differential expression analysis[8], pathway enrichment[9], gene co-expression network analysis in bulk RNA-seq[10], cell annotation[11], cell interaction analysis[12], cell-trajectory inference[13], evaluating cell-state in gene sets, and predicting drug response in scRNA-seq[14]. Many of these approaches rely on open-source algorithms contributed by the research community[15,16].

Nevertheless, the growing diversity and abundance of omics algorithms pose challenges in selecting tools that are accurate, user-friendly, and appropriate for specific analyses. Learning to use various algorithms often leads to computational inefficiencies, as users are

required to adapt to various systems. Moreover, for analyses involving low data quantities, researchers commonly employ web servers and the R language[17], whereas Python is preferred for processing large-scale datasets[18].

Integrating single-cell and bulk sequencing results can be intricate, producing complex, multi-layered data sets that challenge the extraction of meaningful biological insights. A recognized impediment in single-cell sequencing is the "omission"—the omission of certain cell types due to technological constraints on the sequencing platform and interruption of the trajectory of cell differentiation, such as the enzymatic lysis-related loss of podocytes and intercalated cells[19]. For example, the differentiation from hematopoietic cells (HPC) to podocytes was interrupted, and the filtering-induced absence of neutrophils, cardiomyocytes, neuronal cells, and megakaryocytes and the differentiation from neural intermediate progenitor cells (nIPC) to

[1]School of Chemistry and Biological Engineering, University of Science and Technology Beijing, Beijing, China. [2]Daxing Research Institute, University of Science and Technology Beijing, Beijing, China. [3]Center of Precision Medicine and Healthcare, Tsinghua-Berkeley Shenzhen Institute, Shenzhen, Guangdong Province, China. [4]Institute of Biopharmaceutics and Health Engineering, Tsinghua Shenzhen International Graduate School, Shenzhen, Guangdong Province, China. [5]School of Life Sciences, Westlake University, Hangzhou, Zhejiang, China. [6]Academy of Mathematics and Systems Science, Chinese Academy of Sciences, Beijing, China. [7]School of Mathematics and Physics, University of Science and Technology Beijing, Beijing, China. [8]Key Laboratory of Gene Engineering of the Ministry of Education, Institute of Healthy Aging Research, School of Life Sciences, Sun-Yat-Sen University, Guangzhou, Guangdong, China. [9]These authors contributed equally: Zehua Zeng, Yuqing Ma, Lei Hu. ✉e-mail: starlitnightly@gmail.com; cencanxing@ustb.edu.cn; xyyan@mail.sysu.edu.cn; hongwudu@ustb.edu.cn

neurons was interrupted[20,21]. The BD Rhapsody™ single-cell platform overcomes granulocyte loss by accommodating their natural sedimentation[22]. Conversely, bulk RNA-seq of whole tissues intrinsically includes these "omitted" cells. It should be acknowledged that there is no existing algorithm that can directly solve the "omitted" cell problem. However, similar to this problem, there are some deconvolution algorithms, such as TAPE[23], CIBERSORT (CS)[24], MuSiC[25], CIBERSORTx (CSx)[26], and Bisque[27], which are not really effective in solving the "omitted" cell problem because they lack a generative capability. This suggests that Generative Adversarial Networks (GANs) may be the best solution to the "omitted" cell problem.

To address these challenges, we have developed OmicVerse (https://omicverse.readthedocs.io/), a comprehensive Python library designed for transcriptomic research. OmicVerse streamlines access to a spectrum of models and algorithms for bulk-seq and scRNA-seq analyses, improving computational efficiency and visual engagement. Rewritten models and algorithms and integrated different preprocessing options stem from benchmark testing[28] (Supplementary Note 1). Moreover, OmicVerse features single cell trajectory blending from Bulk RNA-seq (BulkTrajBlend), a specialized algorithm for addressing "omission" in single-cell data. BulkTrajBlend employs a beta-variational autoencoder and graph neural network-based algorithm to deconvolve single-cell data from bulk RNA-seq, facilitating the identification of "omitted" cells within the reconstructed single-cell landscape.

## Results

### Design concept of BulkTrajBlend and Benchmarking

The conceptualization of BulkTrajBlend draws upon prior research, proposing that Bulk RNA-seq data is a composite of scRNA-seq data through a nonlinear superposition mechanism[29,30]. Central to this notion is the implementation of the beta-variational autoencoder (β-VAE), a potent tool for approximating Bulk RNA-seq data to scRNA-seq representation[31,32]. Integrating the β-VAE enables the construction of an encoder and decoder from single-cell data, traditionally characterized by unconstrained attributes.

BulkTrajBlend advances the foundational structure of autoencoders (AE) and β-VAE. These enhancements involve (1) employing an AE to construct a Bulk RNA-seq generator analogous to real Bulk RNA-seq inspired by TAPE[23]. We modeled the cellular proportion space of Bulk RNA-seq on the output of the Encoder, the input of the Decoder. Subsequently utilizing ground truth bulk RNA-seq generated from single cell RNA-seq as input of Encoder for calculating the true cellular fractions. (2) When we trained β-VAE using real single cell RNA-seq, the Encoder outputs were V (cell type fraction) and W (cell type correlated generative factor). We added a loss function to minimize the relationship between V and the real cell type fraction. We obtained W for each cell at the end of model training and averaged W for each cell type to represent that cell type. (3) We used the true cell type fraction V calculated by AE with the cell type-associated generating factor W obtained by β-VAE as input to β-VAE for generating single-cell data, and deploying unsupervised clustering to denoise and refine the outcomes of the β-VAE. (4) We employed a graph neural network (GNN) to sample the generated single-cell data, thereby identifying overlapping cell communities. Sampling the overlapping communities of cells helps us to insert "omitted" cells without losing cell continuity.

The methodology based on β-VAE approximates the joint distribution of data **x** and latent generating factors **z** by estimating the probability distribution $q_\theta(\mathbf{z}|\mathbf{x})$ relative to the true posterior $q_\theta(\mathbf{x}|\mathbf{z})$. Here, **x** denotes gene expression data, and **z** characterizes the normally distributed parameters of **x** post-sampling. It is noteworthy that this approximation introduces a level of noise and bias into the generated data (Supplementary Fig. 1e). Consequently, unsupervised clustering is employed as a data refinement strategy to mitigate the impact of noise and enhance data robustness. We use unsupervised

clustering to filter out "noisy" cells, which are identified using community size.

Another notable limitation of β-VAE lies in the unconstrained nature of the decoder's output. This contrasts with the real Bulk environment, where the cellular ratios are not strictly fixed. To address this discrepancy, a simulated Bulk environment is constructed through the sampling of single-cell data, with the procedural details outlined in the "Methods" section. This process is facilitated by a deep neural network (DNN)-based autoencoder model, where the simulated Bulk serves as input, the encoder's output reflects the proportions of actual cells, and the simulated Bulk constitutes the decoder's output. Mean absolute error (MAE) is used as the evaluation metric for both the encoder and decoder. Subsequent to model convergence, the real Bulk data is utilized as input for the AE model, with the critical requirement being the alignment of the generation, based on the best-pretrained decoder, with the real Bulk data. At this point, the cell proportions output by the encoder accurately reflect the cell proportions of the actual Bulk (Fig. 1a).

Given that BulkTrajBlend's primary objective is to interpolate data from original scRNA-seq data, the focus shifts to the targeted extraction of cells from the generated single-cell data. Considering the inherent challenges associated with cell annotation, the input single-cell data containing diverse cell types is expected to exhibit overlaps in real-world scenarios. The "omitted" cells we need to recover should maintain the continuous state of the cells, the traditional community discovery algorithms cannot identify the overlapping cell communities. Cells generated by β-VAE are directly restored to the original single-cell data, which will lose the continuous state of the cell. To solve this problem, we introduce NOCD, a GNN-based algorithm for identifying overlapping communities that achieves the best performance among existing baselines[33]. Utilizing NOCD enables the identification of overlapping cell communities. We also use the "omitted" cell in the overlapping community state as the target cells for recovery. This insight is crucial for the subsequent task of recovering and reconstructing cell differentiation trajectories within the single-cell sequencing data (Fig. 1b).

To assess the efficacy and accuracy of BulkTrajBlend in the context of cell differentiation trajectory recovery, a rigorous benchmarking exercise is undertaken. The VAE module within BulkTrajBlend is systematically compared against alternative generative models, including conditional generative adversarial networks (CGAN)[34] and auxiliary conditional GANs (ACGAN)[35]. This benchmarking exercise involves assessing a range of performance metrics for generating scRNA-seq features and trajectory inference features. These metrics include the correlation of cell-type marker gene expression, marker gene similarity (quantified via cosine similarity), probability of trajectory conversion post-interpolation, and the degree of data variability following interpolation. Notably, the findings consistently demonstrate BulkTrajBlend's superior performance, characterized by heightened correlations in marker gene expression, marker gene similarity, trajectory conversion probabilities, and minimal post-interpolation data variability in the generated single-cell data (Fig. 1c–f, Supplementary Note 2, Supplementary Figs. 2–4).

### Impact of varied hyperparameters on interpolation performance in BulkTrajBlend

This study explores the effect of varying hyperparameter settings on the performance of BulkTrajBlend, a tool reconstruction OPC trajectories in the Dentate gyrus dataset and interpolating Basophil within the HPC dataset. We analyzed the impact of hyperparameter variations by examining five key factors: (1) the number of interpolated cells, (2) the correlation of marker gene expression between interpolated and actual cells, (3) marker gene similarity, (4) transition probabilities following interpolation, and (5) the prevalence of noise clusters.

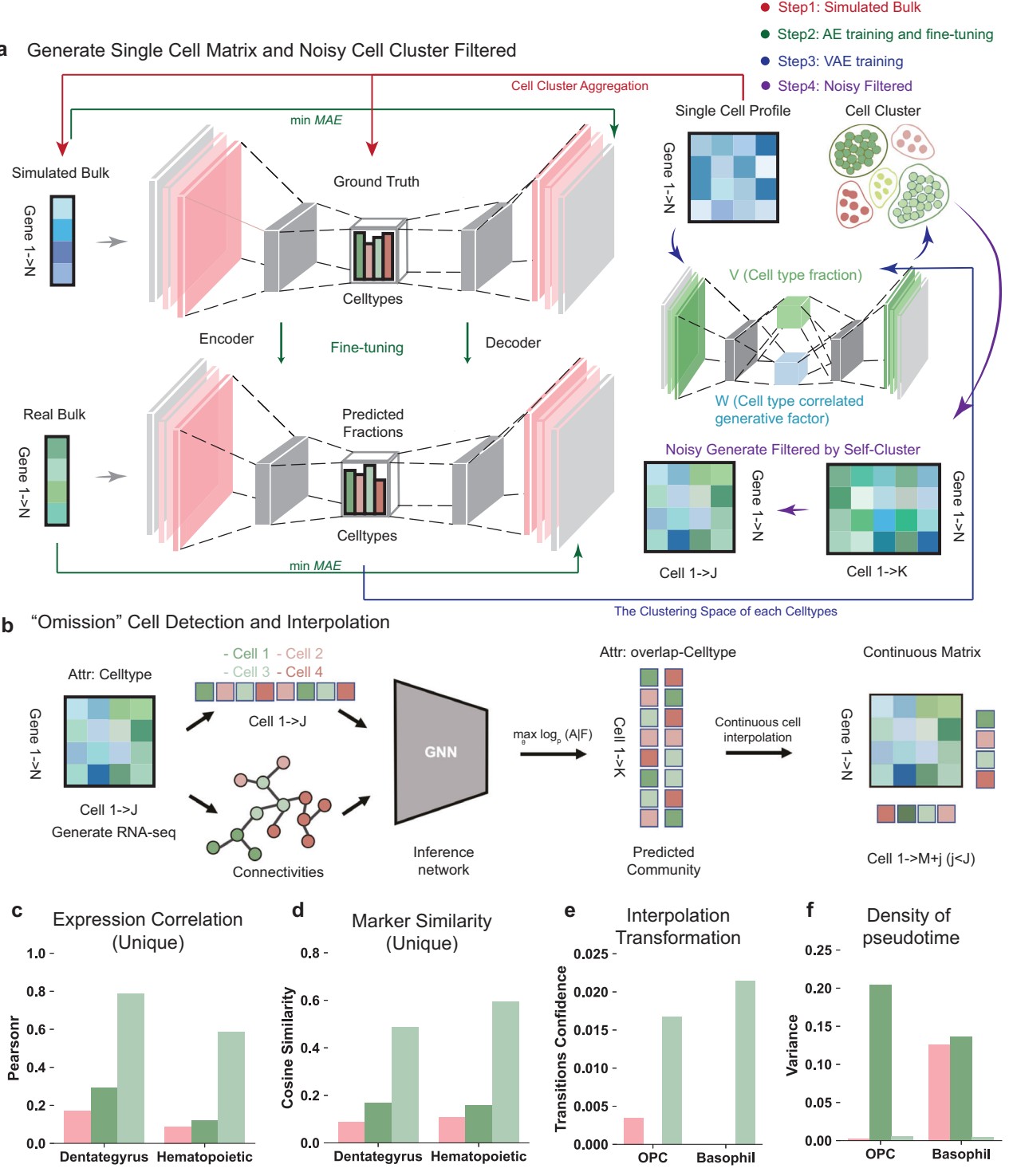

**a**  Generate Single Cell Matrix and Noisy Cell Cluster Filtered

● Step1: Simulated Bulk
● Step2: AE training and fine-tuning
● Step3: VAE training
● Step4: Noisy Filtered

**b**  "Omission" Cell Detection and Interpolation

**c**  Expression Correlation (Unique)

**d**  Marker Similarity (Unique)

**e**  Interpolation Transformation

**f**  Density of pseudotime

■ ACGAN   ■ CGAN   ■ BulkTrajBlend

Initially, the effect of changing the size of the input single-cell data, ranging from 1000 to 20,000 cells, was investigated. An increase in data size resulted in higher correlations of marker gene expression and improved single-cell similarity as performed by BulkTrajBlend (Fig. 2a, b). The transition probabilities, however, were only slightly better (Fig. 2c). Notably, an inverse relationship was found between the saturation of cell numbers and the frequency of noise clusters (Fig. 2d).

Next, the effect of interpolation size was examined, with sizes ranging from 1 to 10 times the original number of target "omitted"

cells. Marker gene correlation and single-cell similarity improved significantly within the 1-4x interpolation range, outperforming the 6-10x range. Conversely, larger interpolation sizes were correlated with a notable increase in noise clusters (Fig. 2e–h).

Contrary to expectations, a detailed analysis of the number of neurons in BulkTrajBlend's hidden layer, with a range from 64 to 1024, revealed that a hidden layer with only 64 neurons exhibited the highest marker gene correlation, similarity, and transition probability for interpolated single cells, while also reducing noise cluster occurrences (Fig. 2i–l).

**Fig. 1 | Architecture of the BulkTrajBlend framework. a** Single-Cell Profile Generation in BulkTrajBlend: This stage outlines the creation of single-cell profiles. An initial single-cell profile, representing the ground truth for cell fractions, and simulated bulk transcriptome data are input into an autoencoder (AE). Simultaneously, real bulk transcriptome data serve as the optimal input for the AE. The AE's predicted cell fractions define the clustering space of the resulting single-cell profile, which is then processed by a β-VAE to generate a profile similar to that of real bulk data. Any noise in this profile is reduced using unsupervised clustering. **b** "Omitted" Cell Detection in BulkTrajBlend: Here, a neighborhood graph constructed via UMAP based on the generated single-cell data identifies nodes corresponding to individual cells and delineates distinct communities by cell type. The annotated graph is the input for a Graph Neural Network (GNN) that detects overlapping communities and identifies mixed cell types, which are then reintegrated into the original single-cell profile. Overlap-CellType: a one-hot matrix of

cell types for the overlapping Celltypes. **c** Correlation Score of Cell-Type Marker Gene Expression: This component displays correlation scores for cell-type marker gene expression across three models within the Dentate Gyrus and Hematopoietic datasets. **d** Cell-Type Marker Similarity Assessment Using Cosine Similarity: This part addresses the assessment of similarities between cell-type marker genes using cosine similarity. **e** Probability of Cell Conversion: The framework evaluates the likelihood of nIPC (neurogenic intermediate progenitor cells) becoming OPC (oligodendrocyte progenitor cells) against the backdrop of interpolated OPC cells in the Dentate Gyrus dataset, and the corresponding likelihood for the conversion of HSC (hematopoietic stem cells) to Basophil cells with interpolated Basophil cells in the Hematopoietic dataset. **f** Pseudotime Density for OPC Cells: This final component illustrates the pseudotime density of OPC cells incorporating interpolated OPC cells in the Dentate Gyrus dataset, coupled with an analogous representation for Basophil cells post-interpolation in the Hematopoietic dataset.

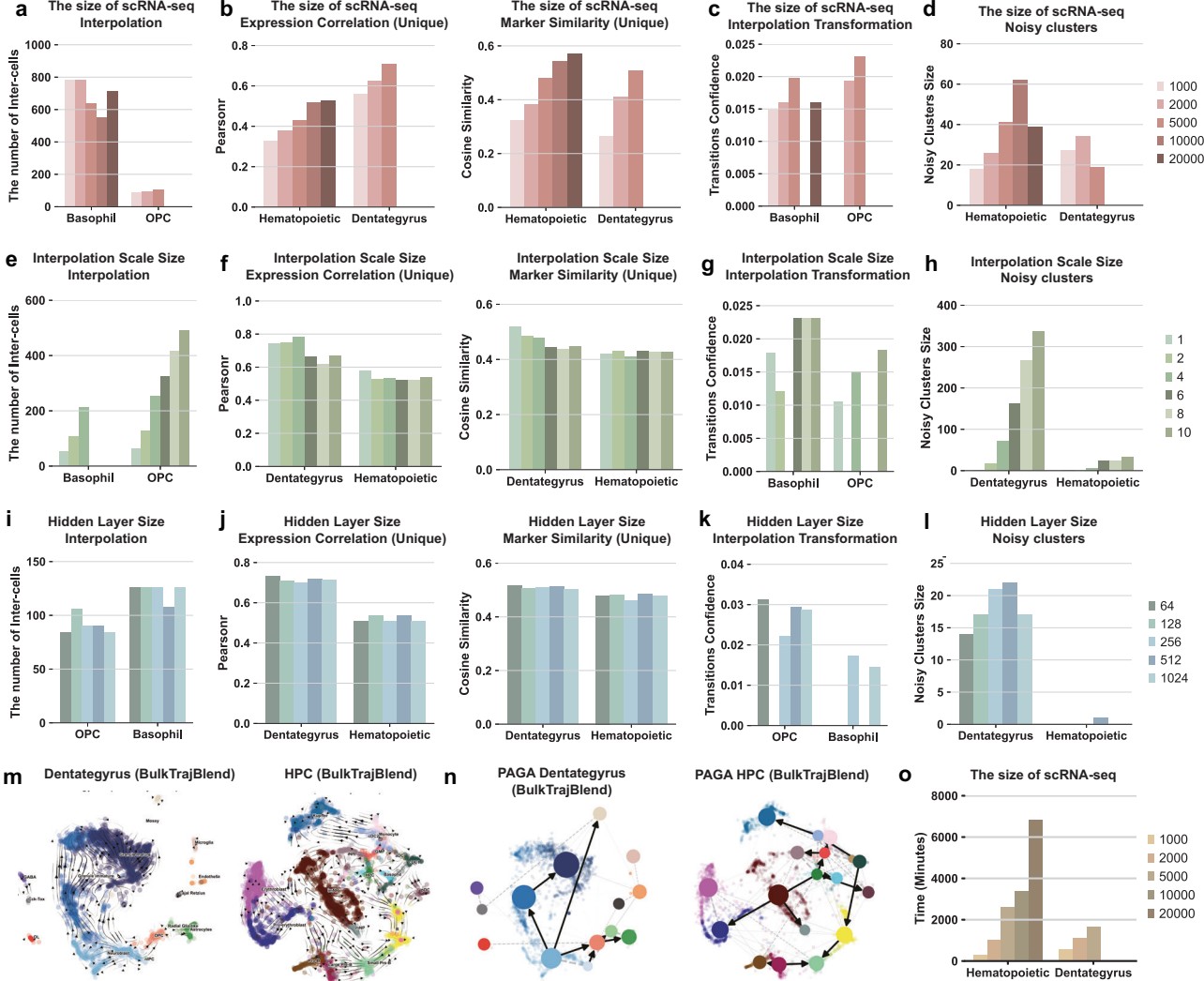

**Fig. 2 | The systematic hyperparameter testing for interpolation performance.** The tests examine varying sizes of raw single-cell profiles as input in **a–d**: (**a**) The quantity of "omitted" cells generated from Basophil cells in the Hematopoietic dataset and OPC cells in the Dentate Gyrus dataset, respectively. **b** The analysis juxtaposes two aspects: on the left panel, the expression trends' correlation of marker genes between the reference and generated single-cell profiles; and on the right panel, the similarity between marker genes of the two profiles. **c** The transition probability of the generated target cells is computed along the cellular developmental trajectories, with Basophil cells in the Hematopoietic dataset and OPC cells in the Dentate Gyrus dataset. **d** The extent of noise clusters present in

single-cell profiles, with the Hematopoietic dataset on the left and Dentate Gyrus on the right. **e–h** The scale sizes of the generated target cells utilized as input is scrutinized. **i–l** The sizes of neurons in the hidden layer vary as input. **m** The flow trend of cell developmental trajectories of neurogenic intermediate progenitor cells (nIPC) is visualized on UMAP plots for the Dentate Gyrus on the left and the Hematopoietic dataset on the right. **n** Cell-state transition directed graphs within the trajectory of Partition-based Graph Abstraction (PAGA) graphs are presented for the Dentate Gyrus on the left and Hematopoietic dataset on the right. **o** The model's runtime in relation to different sizes of raw single-cell profile inputs is illustrated.

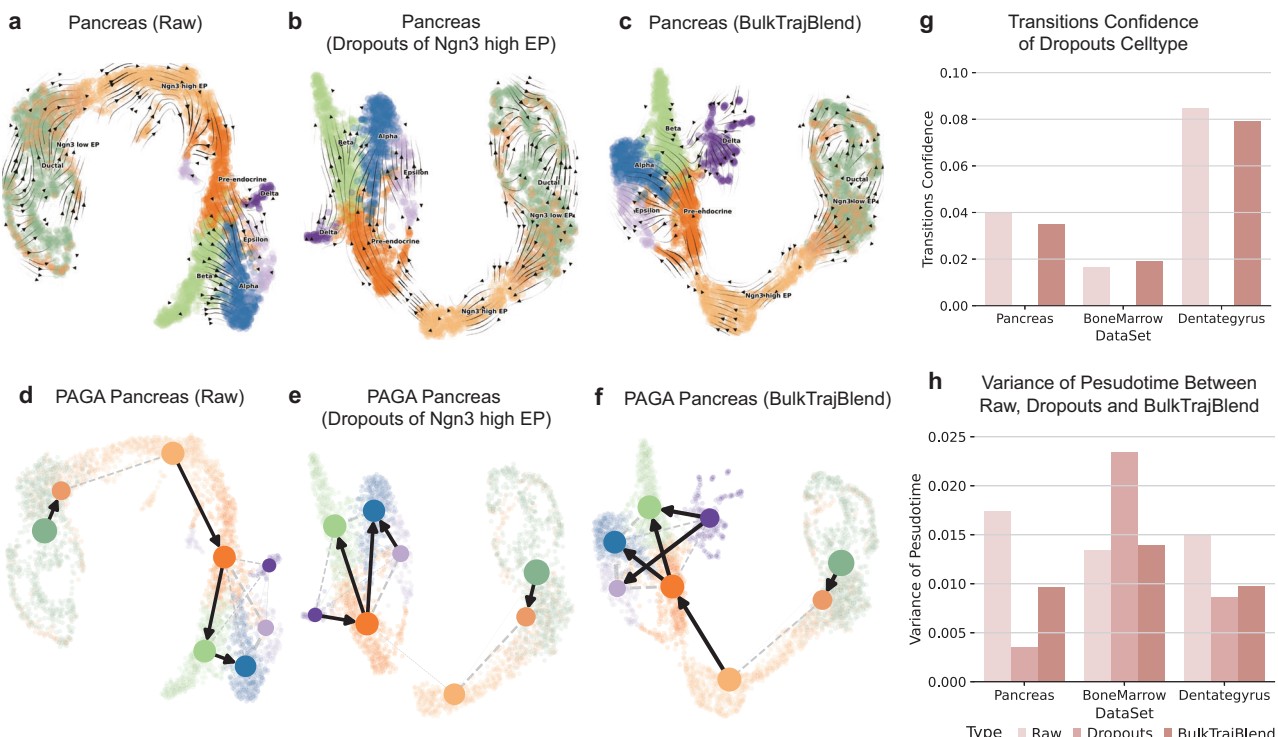

**Fig. 3 | Reconstruction of cell developmental trajectories in simulated "omission" within single-cell Profiles. a–c** Sequentially depicted are the raw pancreas dataset's velocity stream, the effect of simulated omission via cell dropouts, and the refined dataset post-interpolation with BulkTrajBlend for dropout imputation as determined by pyVIA. The UMAP embedding is color-coded by cell type, consistent with the initial cluster annotations. Explained are the following cell types: Ngn3$^{High}$ EP, Ngn3$^{High}$ endocrine progenitor-precursor; Ngn3$^{Low}$ EP, Ngn3$^{Low}$ endocrine progenitor-precursor, Alpha, glucagon- producing α-cells; Beta, insulin-producing β-cells; Delta, somatostatin-producing δ-cells and Epsilon, ghrelin-producing ε-cells. **d–f** Displayed in sequence is the directed graph overlaid on the UMAP embeddings for the raw pancreas dataset, the dataset with "omission" in cell dropouts, and the dataset post-BulkTrajBlend interpolation based on pyVIA's dropout assessments. **g** The confidence in cell state transitions as determined by pyVIA is presented for various datasets and experimental conditions. The corresponding color bars signify the methodology employed. Specifically, for the pancreas dataset with Ngn3$^{High}$ EP dropouts, the displayed confidence indicates the transition from Ngn3$^{High}$ EP to pre-endocrine cells. In the bone marrow dataset with HSC dropouts, the values represent the transition confidence from HSC to Monocytes. Likewise, the Dentate Gyrus dataset with dropouts of Granule Immature cells indicates the transition confidence from Granule immature to Granule mature cells. **h** The variance in pseudotime, as estimated by pyVIA, is documented across different datasets and experimental manipulations.

In conclusion, the ideal hyperparameter setting involves using the entire single-cell dataset, interpolating at a scale of 2x or 4x, and configuring a hidden layer with 64 neurons. Under these optimal hyperparameters, BulkTrajBlend effectively reconstructs the nIPC-OPC developmental flow pattern in dentate gyrus datasets and the HSC-Basophil flow pattern in hematopoietic system development datasets (Fig. 2m,n). It is important to note that using the full single-cell dataset improves accuracy, which also significantly increases computational demands (Fig. 2o).

### Proficient reconstruction of cell developmental trajectories in simulated "omission" single-cell profiles

Our study extended beyond evaluating BulkTrajBlend's ability to reconstruct developmental trajectories in real datasets, by also examining its performance within simulated datasets. We crafted three simulated datasets with specific "omission": the first omitted a subset of Ngn3$^{High}$ endocrine progenitor-precursor (Ngn3$^{High}$ EP) cells in mouse pancreas development, the second removed immature granules from mouse dentate gyrus neurons development, and the third excluded hematopoietic stem cells (HSC) mesomorphic cells from human bone marrow development. These cells were successfully recognized in the reconstructed developmental trajectories within these simulated "omission" datasets (Fig. 3a–c, Supplementary Fig. 5a–c, Supplementary Fig. 5g–i).

In the mouse pancreatic development dataset, PAGA plots illustrated a baseline probability of 0.04 for Ngn3$^{High}$ EP cells

differentiating into Pre-endocrine cells. In the corresponding "omission" dataset, this probability was 0. BulkTrajBlend interpolation increased the probability to 0.035 (Fig. 3d–g, Supplementary Fig. 6a–c). In the mouse dentate gyrus neurons development, Granule Immature cells had baseline differentiation probability to Granule Mature cells of 0.018, while no probability was observed in simulated "omission" dataset. BulkTrajBlend's interpolation resulted in a probability increase to 0.019 (Fig. 3g, Supplementary Fig. 5d–f, Supplementary Fig. 6d–f). In human bone marrow development, hematopoietic stem cells stage 2 (HSC 2) cells showed a differentiation probability into monocytes of 0.082, compared to 0 in the simulated "omission" dataset. Following BulkTrajBlend interpolation, the probability increase to 0.079 (Fig. 3g, Supplementary Fig. 5j–l, Supplementary Fig. 6g–i). Notably, the original pseudotime variability in the three datasets was preserved after interpolation (Supplement Note 3). These analyses collectively highlight BulkTrajBlend's effectiveness in accurately reconstructing authentic developmental trajectories.

### OmicVerse provides a comprehensive analysis platform for Bulk RNA-seq data

Bulk RNA-seq is an established method for investigating the transcriptome of combined cellular samples, tissue or biopsies[6]. It probes gene expression, isoform variations, alternative splicing, and single-nucleotide polymorphisms, revealing critical biological information such as copy number variations, microbial contamination, transposable elements, cell-types deconvolution, and neoantigens. Advances

in bioinformatics have enhanced the ability to reveal these hidden dimensions in Bulk RNA-seq data, expanding its analytical applications.

OmicVerse integrates an extensive collection of Bulk RNA-seq analysis algorithms, previously developed mostly in R but now increasingly in Python, to promote their utilization and interconnectivity[36]. Our integration enhances the existing repertoire of analysis algorithms catering to single-cell, spatial transcriptomics, as well as machine learning and deep learning models[37].

The platform hosts a comprehensive assortment of Bulk RNA-seq algorithms, including pyComBat[38] for batch correction, pyDEG for differential expression analysis using Deseq2[39], t-test, and Wilcoxon tests, pyPPI for protein-protein interaction using STRING web API[40], pyWGCNA for gene co-expression network[41], pyGSEA for gene set enrichment analysis[42], and pyTCGA for The Cancer Genome Atlas (TCGA) data analysis, complete with survival analysis (Fig. 4a).

To evaluate the OmicVerse's analytical pipeline, we analyzed Alzheimer's disease (AD) data, beginning with pyDEG to identify differential expressed genes between AD patients and controls, highlighting the top 10 foldchange genes. Then, we conducted Gene Set Enrichment Analysis at the gene level using pyGSEA, ordering genes according to p-values derived from pyDEG's differential expression analysis. We further built a co-expression network from the top 5000 genes exhibiting the highest absolute median difference (MAE), selecting the most differential expression module for visualization (See Supplementary Note 4 for Methods).

OmicVerse's workflow simplifies Bulk RNA-seq analyses with minimal coding required (Fig. 4b). Parameter adjustments may enhance visual outputs. Our analysis revealed 56 genes differentially expressed in AD: 48 upregulated and 8 downregulated. Box plots showcased the most altered genes (Fig. 4c–e). Gene Set Enrichment Analysis exposed over-represented pathways relevant to Alzheimer's, consistent with established literature (Fig. 4f,g). Moreover, we focused on the most variable genes from the top 5000, discerning 12 modules through pyWGCNA at 5 soft threshold. Notably, modules 4 and 5 showed the highest rates of differential gene expression, with module 5 containing APP proteins. Further probing of these modules provides insight into their network connectivity (Fig. 4h–j).

## OmicVerse provides a versatile multifaceted framework for Single-Cell RNA-Seq Analysis

Single-cell RNA-seq is a powerful high-throughput technique that enables the measurement of gene expression patterns and cell types at the single-cell level. It has become a crucial technique for delineating cellular heterogeneity, differentiation, and disease mechanisms, particularly in cancer research. scRNA-seq unravels tumor cell diversity and tracks tumor progression to anticipate cellular deterioration[43]. The breadth of scRNA-seq data analysis facilitated by OmicVerse includes cell annotation, examination of cell interactions, trajectory inference, states evaluation within gene sets, and drug responses prediction[44]. The framework supports Anndata-standardized data processing for integrated downstream analysis and benefits from benchmarked data transformations[28]. Preprocessing methods in OmicVerse feature optimal logarithmic transformation with pseudo-count addition, principal-component analysis (PCA), and Pearson residual normalization. For visualizing reduced dimensions, it employs GPU-accelerated Uniform Manifold Approximation and Projection (UMAP) through pymde[45].

Incorporating a suite of state-of-the-art scRNA-seq algorithms, OmicVerse's integrated toolset includes pyHarmony[46], pyCombat[38], scanorama[47] for batch correction, pySCSA[48], updated with CellMarker[49] 2.0 and CancerSEA[50] for enhanced cell-type annotation, CellPhoneDB[12] for cell-cell interactions analysis pyVIA[13] for trajectory inference, AUCell for geneset score evaluates based on Area Under the Curve[51], and scDrug for drug prediction[14] (Fig. 5a). The OmicVerse framework also introduces SEACells for metacell analysis, effectively minimizing single cell profile noise[52]. Importantly, the data format input for all the aforementioned methods is consistent, enabling users to conduct analyses using Anndata format, with significantly improved visualization for more elegant results. OmicVerse's user-friendly nature and straightforward application are exemplified in Fig. 5b.

Illustrating Omicverse's practical application in scRNA-seq, we analyzed a colorectal cancer (CRC) dataset, emphasizing the tumor microenvironment (TME) cell atlas integration[53,54]. Beginning with automatic cell annotation via pySCSA, the results showed high concordance with manual annotations (Fig. 5c), with an f1_score of 0.856, highlighting OmicVerse's annotation accuracy (Fig. 5d). Using AUCell, we confirmed the expected signaling pathway enrichment in cell-specific receptor pathways: the B-cell receptor signaling pathway was prominence in B cells, while the T-cell receptor signaling pathway was most pronounced in T cells and NK cells (Fig. 5e). In addressing the sparsity inherent in previous CRC single-cell data analysis and to enhance resolution and depth, we utilized SEACells to extract metacells from the scRNA-seq data. After 39 epochs, the metacell aggregation iteration converged, attaining a high cell purity of 0.98, with compactness and separation values closely approximating 0 (Fig. 5f, Supplementary Fig. 7a–c). The SEACells algorithm enhanced cell type differentiation, with the signal intensity for receptor pathways being significantly accentuated (Supplementary Fig. 7d).

Furthermore, we traced epithelial-to-cancer cell differentiation trajectories using pyVIA and annotated cancer cell types within the epithelial population with pySCSA, identifying distinct pathways including Epithelial-to-Mesenchymal Transition (EMT) and Metastasis. This analysis provided deep insights into cancer progression (Fig. 5g). By commencing the trajectory with stemness as the starting point, we delineated the pseudotime trajectory of cancer cell differentiation, revealing three distinctive directions: EMT-Differentiation and Metastasis, representing two stages in the transition from epithelial cells to cancer cells. This analysis provided deep insights into the dynamics of cancer evolution. In a parallel approach, metacells within the epithelial cell subpopulation were subjected to further aggregative analysis. Due to the inherent similarities among epithelial cells, the average cell purity of the metacells obtained was reduced to 0.9, while compactness and separation values remained in close proximity to 0 (Supplementary Fig. 7e, f). Consequently, we extrapolated the metacells of epithelial cells into trajectories, revealing that EMT-differentiation and Metastasis served as the two primary differentiation pathways, aligning with the analysis conducted on all cells (Supplementary Fig. 7g–i).

Finally, to investigate the interaction network between epithelial cells and other TME cells, we established a CRC cell communication network using CellPhoneDB (Fig. 5h). The analysis included immune cells, including B-cells, T-cells, NK-cells, and plasma cells, exploring their interactions with eight subtypes of epithelial cells. The analysis revealed that PPIA-BSG and LTB-LTBR were recurrent ligand-receptor pairs mediating the recognition of cancer epithelial cells by immune cells (Fig. 5i). Notably, PPIA-BSG and LTB-LTBR have been linked to a positive correlation in various cancers and are associated with poor prognosis[55,56]. OmicVerse's data harmonization significantly streamlines this comprehensive analysis, enabling researchers to delve into personalized explorations as outlined in our detailed tutorial (Refer to Supplementary Note 5 for the Methods).

## OmicVerse performed multi-omics analysis with MOFA and GLUE

Single-cell sequencing advancements enable the investigation of biological systems across different tissue levels. A key element in scRNA-seq is understanding the impact of chromatin accessibility variation, which is quantified by Single-cell sequencing assay for transposase-accessible chromatin (scATAC-seq). The conjoined analysis of scATAC-seq and scRNA-seq data is critical for unraveling transcriptional

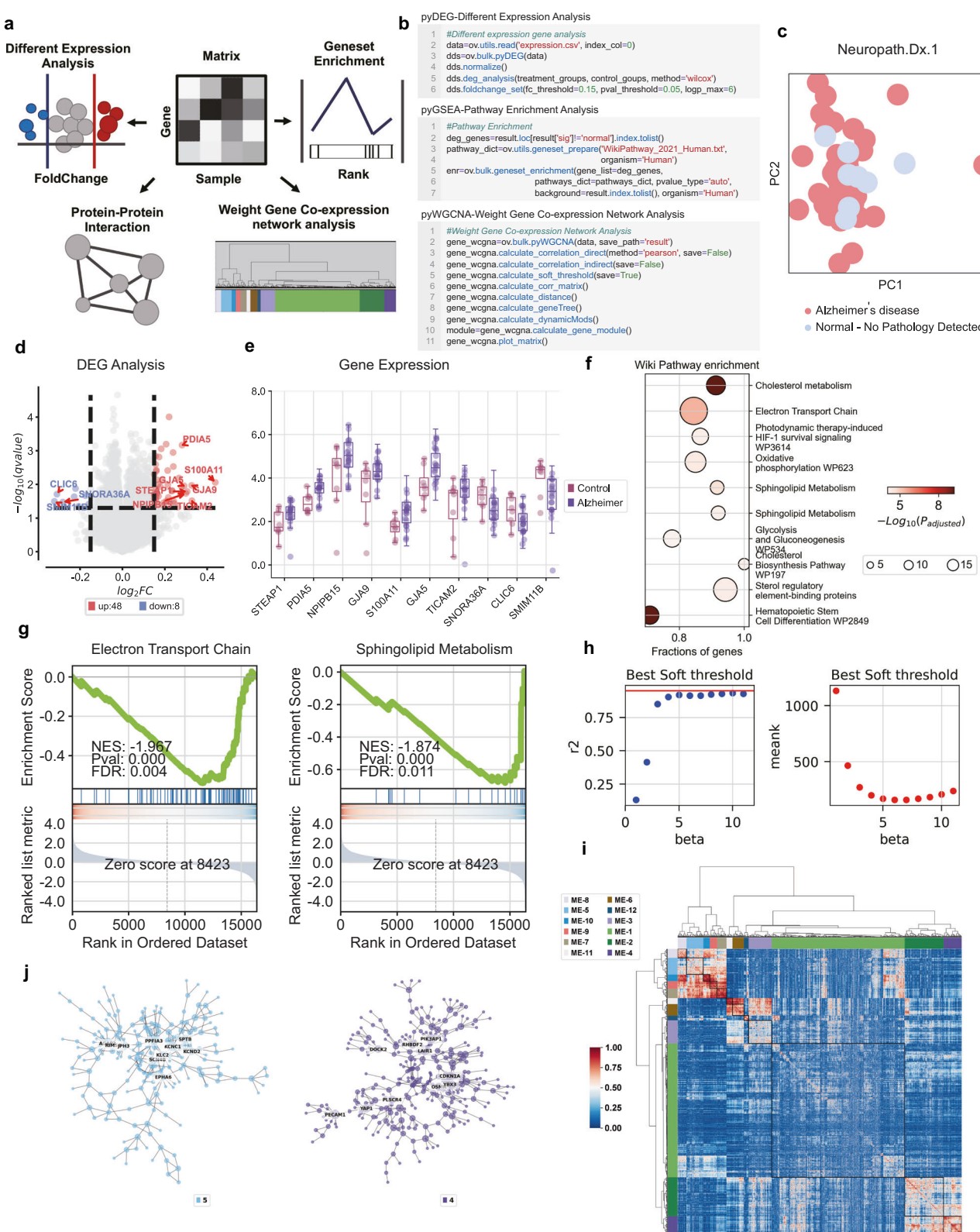

regulatory complexities. While scNMT-seq can capture both modalities simulaneously, obtaining unpaired data from identical tissues is more common[57]. Addressing this disparity, Graphical Linkage Unified Embedding (GLUE) offers a Graphical Linkage Unified Embedding solution for integrating unpaired data[58], and Multi-Omics Factor Analysis (MOFA) elucidates the variations within omics data[59]. OmicVerse utilizes both GLUE and MOFA to reveal transcriptional regulatory dynamics.

Within OmicVerse, the GLUE_pair algorithm leverages the Pearson correlation coefficient to compute cell similarity between scRNA-seq and scATAC-seq base on embedding from GLUE (Fig. 6a). The accuracy of GLUE_pair is verified using the Adjusted Rand Score (ARI) to confirm cell type congruence post-normalization. For the analysis of paired cell modalities, OmicVerse applies MOFA's core algorithm, simplifying ensuing data analysis and visualization tasks (Fig. 6a), all achievable with minimal coding (Fig. 6b).

**Fig. 4 | A comprehensive overview of Bulk RNA-seq data analysis utilizing OmicVerse. a** A graphical depiction illustrates various analyses: differential expression analysis (pyDEG), gene set enrichment analysis (pyGSEA), protein-protein interaction analysis (pyPPI), and weighted gene co-expression network analysis (pyWGCNA). **b** A code snippet demonstrates how to import data and execute pyDEG, pyGSEA, and pyWGCNA, incorporating continuous covariates. **c** Principal Component Analysis (PCA) embeddings distinguish samples within Alzheimer's and control groups. **d** A volcano plot highlights differentially expressed genes; those upregulated are marked in red, while downregulated genes are indicated in blue. **e** A box plot reveals the top 10 genes with the most significant fold change between Alzheimer's ($n = 44$) and control groups ($n = 46$) (Boxplot is displayed with the center-line as median, the box limits as lower and upper quartiles, and with whiskers covering the most extreme values within 1.5 x Interquartile-Range). **f** WikiPathways enrichment results are visualized, with dot size correlating to the gene count for each function and color intensity reflecting $p$-value significance – darker hues indicate higher pathway enrichment (top10 of positive and negative NES, padj < 0.05, padj calculated by GSEApy python package, For the statistical analysis, we used the $p$value adjustment to control for multiple comparisons. The enrichment scores were evaluated as two-sided, considering both positive and negative deviations from the expected distribution under the null hypothesis). **g** Gene set enrichment analysis (GSEA) is executed using WikiPathways gene sets, with enrichment scores and $p$-values derived from a weighted two-sided Kolmogorov–Smirnov-like statistic and normalized for gene set size, producing the Normalized Enrichment Score (NES). **h** The optimal soft threshold is determined, where the horizontal axis represents the soft threshold gradient, the left vertical axis corresponds to the scale-free fit index (with higher values preferred), and the right vertical axis reflects the average node connectivity (with lower values preferred). **i** A gene clustering dendrogram illustrates dissimilarity based on topological overlap, combined with module color assignments. Consequently, twelve co-expression modules are identified, each displayed in a distinct color. An accompanying heatmap depicts the correlation among the 5000 genes within each module. **j** Modules 4 and 5, which are scale-free networks, are shown where each node represents a gene. The node size corresponds to gene connectivity, and color denotes the module affiliation, with the five most central genes in each module labeled.

Demonstrating the integration of GLUE and MOFA, we analyzed simultaneous single-nucleus RNA-seq (snRNA-seq) and single-nucleus ATAC-seq (snATAC-seq) data from cortical regions of Alzheimer's disease patients[60]. Our analysis of aligned cell types uncovered consistent patterns indicative of common cellular states (Fig. 6c,d). From a random subset of 5000 paired cells, MOFA unveiled 13 factors (Fig. 6e,f). The factors 1-6 accounted for RNA-related variance, while the second for ATAC-related variance. The interaction among these factors and cell types revealed significant associations: EX-signature with Factor 1, PER.END-signature with Factor 5, ASC-signature with Factor 2, MG-signature with Factor 3, and INH-signature jointly detailed by Factors 6 and 4. Additionally, gene weights for each factor uncovered genes with the most considerable influence on their respective signatures (Refer to Supplementary Note 6 for the Methods, Supplementary Fig. 8a–c).

## Discussion

The innovative fusion of the variational autoencoder and graph neural networks combined in the creation of the BulkTrajBlend framework. This framework aims to deconvolve scRNA-seq data within Bulk RNA-seq and elucidate precise cell-specific developmental trajectories in scRNA-seq. It demonstrates significant accuracy and robustness, due in large part to the unique integration of the topological overlap community in graph neural networks, which skillfully addresses the potential bias introduced by unsupervised clustering in the single-cell data outcomes.

A conceptual parallel exists between back-calculating cell proportions in Bulk RNA-seq from scRNA-seq and using Bulk RNA-seq as a scaffold for interpolating scRNA-seq. However, the latter is inherently more challenging due to the need to accurately interpolat the inadequate target cell type. While numerous single-cell generators perform well in generating scRNA-seq data, the incorporation of unknown information remains an intrinsic challenge. For example, scDesign3 is a proficient statistical simulator that creates realistic single-cell data by learning interpretable parameters from actual scRNA-seq data. Nevertheless, reconstructing cell developmental trajectories often requires elusive parameters, which necessarily leverages known data from Bulk RNA-seq[61]. Hence, BulkTrajBlend is meticulously crafted based on the principles of scDesign3[61] and scGen[32], with the state space and parameters being informed by Bulk RNA-seq. Notably, cell categorization in the resulting single-cell data often relies on unsupervised annotation. By introducing GNN, BulkTrajBlend effectively reduces resolution-dependent issues associated with unsupervised clustering.

While BulkTrajBlend can efficiently extract the state space of cells from Bulk RNA-seq and interpolate the original scRNA-seq data, this interpolation relies on the selection of the reference scRNA-seq versus the reference Bulk RNA-seq data. We suggest that users can adopt an additional comprehensive single-cell profile to train BulkTrajBlend and then perform interpolation of their data, thereby avoiding generating BulkTrajBlend without information about the target cells.

Upon devising the interpolation algorithm for Bulk RNA-seq in scRNA-seq, it became apparent that a unified Python-based framework for comprehensive dual analysis of these platforms was missing. To fill this void, we developed OmicVerse, seamlessly integrating single-seq and bulk-seq. OmicVerse introduces a specialized analysis object for each omics layer, facilitating streamlined analysis and ensuring an intuitive user experience. OmicVerse not only has a well-established scRNA-seq ecosystem like Seurat, which complements Scanpy, but also features a unique Bulk RNA-seq ecosystem, thus offering a consistent and user-friendly interface (Supplement Note 7).

As an integrated framework for both Bulk and single-cell RNA-seq analysis, OmicVerse offers a suite of analytical tools that include, but are not limited to:

(1) **Bulk RNA-seq**: OmicVerse provides comprehensive functionalities, including multi-sample integration, batch effect correction, differential gene expression analysis, gene set enrichment analysis, protein interaction network construction, the identification of gene co-expression modules, and TCGA database preprocessing.

(2) **Single-cell RNA-seq**: OmicVerse offers robust features, including multi-sample quality control, batch effect removal and integration, automated cell type annotation (with multiple databases support) and migration annotation, cell type and gene set enrichment analysis, developmental trajectory reconstruction, metacell identification, cellular interaction network analysis, and drug response prediction. It also covers scATAC-seq integration and multi-omics analysis, inherently linked to RNA-seq.

(3) **Bulk RNA-seq to scRNA-seq**: OmicVerse enhances the deconvolution of Bulk RNA-seq, cell proportions estimation, interpolation the scRNA-seq data and the recovery of developmental trajectories within scRNA-seq. Acting as a critical bridge in the transition from Bulk to single-cell RNA-seq.

The OmicVerse documentation provides a detailed Application Programming Interface (API) reference for each algorithm, coupled with tutorials that clarify their functions, limitations, and synergies with other bulk and single-seq analysis tools. These resources are accessible via Google Colab, offering a free computational workspace for pipeline examinations. OmicVerse also has comprehensive developer documentation that makes it easy for users to add tools to the ecosystem following a consistent development logic.

Our primary goal was to foster an ecosystem replete with visually engaging and insightful visualizations, fully integrated

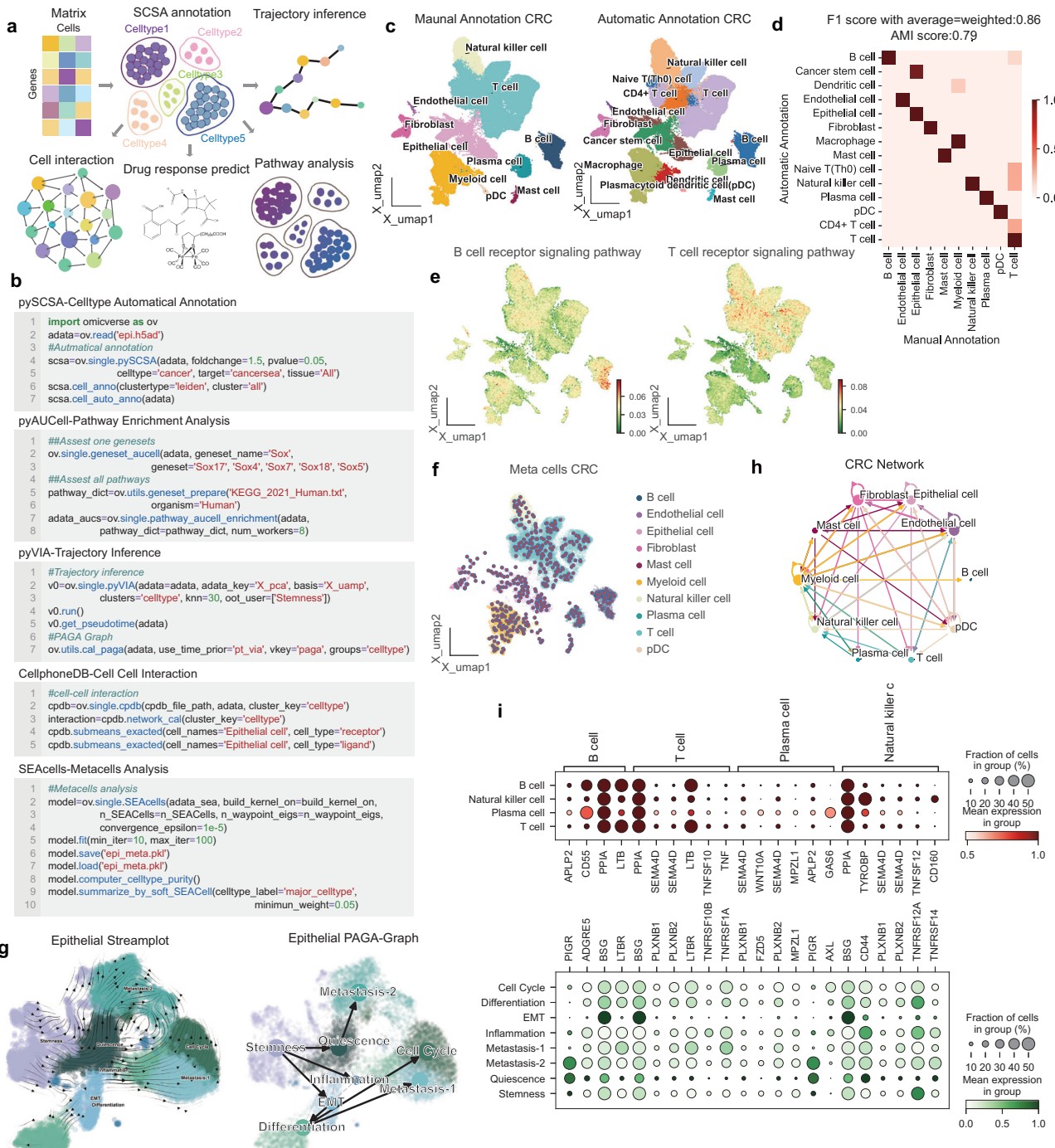

**Fig. 5 | OmicVerse a comprehensive analytical platform for single-cell RNA-seq analysis. a** A graphical overview highlights crucial analysis modules: cell type annotation (pySCSA), cellular interactions (CellPhoneDB), trajectory inference (pyVIA), pathway analysis (AUCell), and drug response prediction (scDrug). **b** An example code snippet illustrates the process for loading data and conducting analyses using pySCSA, CellPhoneDB, pyVIA, AUCell, and SEACells, with the inclusion of continuous covariates. **c** UMAP plot visualizes single-cell RNA sequencing (scRNA-seq) data from colorectal cancer (CRC) patients. The plot contrasts manual cell type annotations, shown in the left panel, with automatic annotations depicted in the right panel. **d** The concordance between manual and pySCSA-generated annotations is presented in a row-normalized confusion matrix. **e** Pathway

enrichment within CRC cells is elucidated in a UMAP visualization, with the left side indicating B cell receptor signaling and the right side detailing T cell receptor signaling, as analyzed by AUCell. **f** Metacell composition within the CRC dataset is revealed in a UMAP plot. **g** Epithelial cell subpopulations in CRC are displayed in a UMAP plot; automated annotations by pySCSA are demonstrated on the left, complemented by a cell state transition directed graph derived from a Partition-based Graph Abstraction (PAGA) trajectory on the right. **h** CellPhoneDB computes an interaction network between CRC cell types, offering insights into intercellular communication. **i** Scaled mean expression levels of genes that code for interacting ligand-receptor proteins, identified by CellPhoneDB, are shown in dot plots to underscore the supporting interactions between immune and epithelial cells.

within the Python programming environment. OmicVerse allows users to perform extensive transcriptome analysis using a single programming language, tapping into the collective machine-learning knowledge and models available within the Python

community. We anticipate that OmicVerse will continue to grow, with updates introducing additional algorithms, features, and models. Ultimately, OmicVerse aims to act as a driving force for the bulk and single-seq community, encouraging the prototyping of

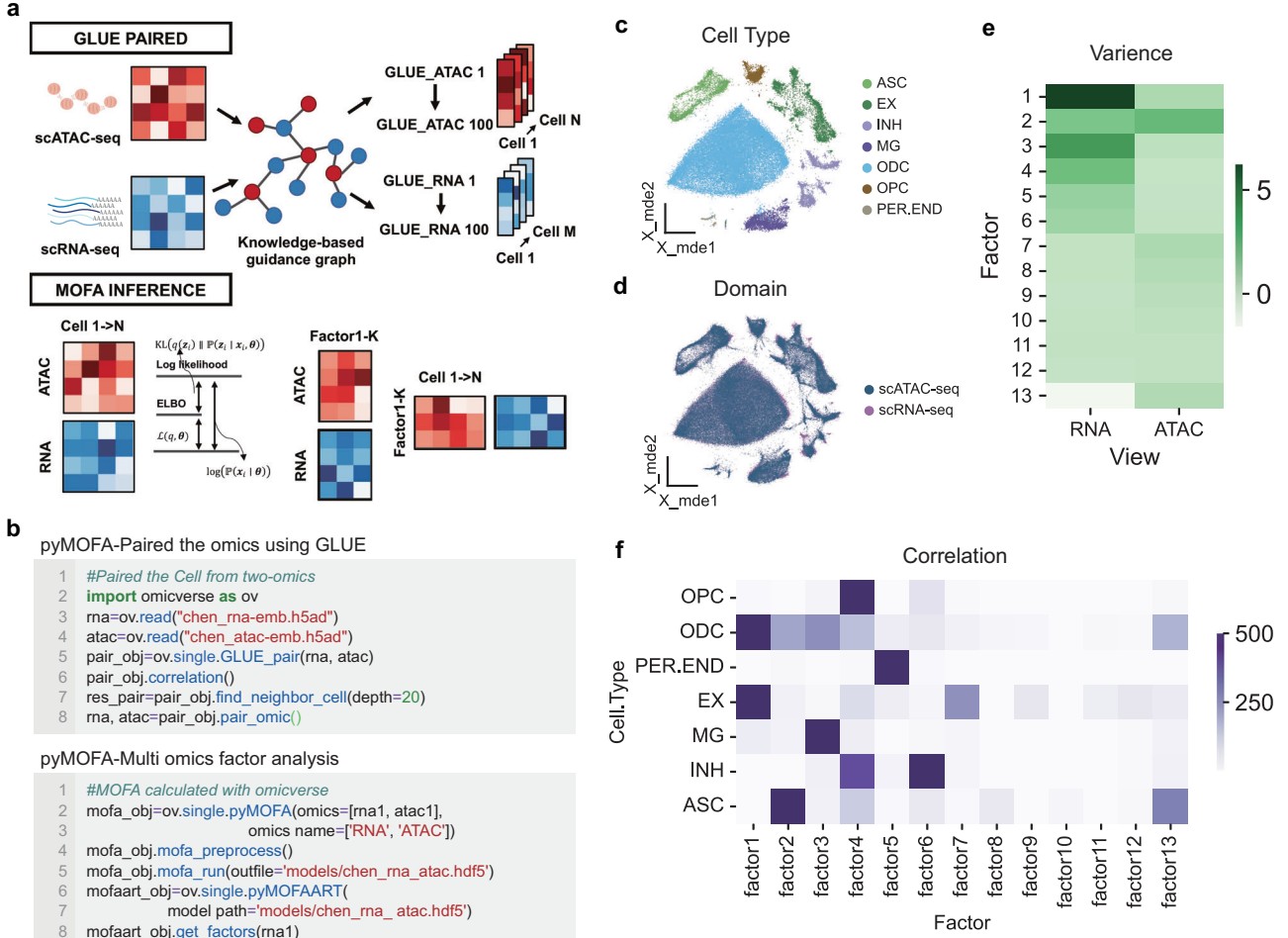

**Fig. 6 | The integration of multi-omics data analysis by OmicVerse, utilizing both MOFA and GLUE. a** The representation includes a graphical model of cell type correlations using GLUE, alongside an illustration of cell variance captured by MOFA, as indicated by the Evidence Lower Bound (ELBO). **b** A sample code snippet is provided for the import and processing of data via pyMOFA tools. **c** A UMAP plot shows the distribution of cell types identified in scRNA-seq data from patients with Alzheimer's Disease. **d** Integrated cell embeddings from various omics layers are displayed in UMAP visualizations, with color-coding reflecting the respective omic strata. **e** A heatmap illustrates the percentage of variance accounted for by each factor (displayed as rows) across different omics layers. **f** Another heatmap exhibits the results of correlation analyses between cell types and the MOFA factors. Colors represent tests of significance.

---

various models, establishing standards for RNA-omics analysis, and expanding the potential for scientific exploration.

## Methods
### Methods for BulkTrajBlend
BulkTrajBlend is primarily designed to address the issue of "omitted" cells in single-cell data, making the inference of developmental or differentiation trajectories continuous. To achieve this goal, we designed BulkTrajBlend to generate potential "missing" cells from bulk RNA-seq data for inferring pseudo-time cell trajectories. This process consists of the following four steps (where communities represent cell types):

**Cell proportion calculation.** To estimate the proportion of cells in Bulk RNA-seq, we first annotated the single-cell data with respective cell types and aggregate the gene counts of single cells by cell type, resulting in an $N*M$ matrix, where $M$ represents the number of cell types and $N$ represents the number of genes. We define this $N \times M$ matrix as the simulated Bulk RNA-seq cell type matrix, and then we sum $M$ columns of each row to get the simulated Bulk RNA-seq $B_{simulated}$, and we input the simulated Bulk RNA-seq into the self-encoder of AE. In the self-encoder, we define the output of the encoder as $T$, and we make $T$ close to $\frac{Number\ of\ the\ cell}{Number\ of\ all\ cells}$, i.e., Cell

Proportion, by training AE. We then define the output of the generator as $G$ and we make $G$ and $B_{simulated}$ close to each other by MAE as an evaluation. After training the optimal AE, we change the input to real Bulk RNA-seq $B_{groundtruth}$, at which time the output of the encoder, $T$, is the Cell Proportion corresponding to real Bulk, which we use as the range of the generator space for the subsequent β-VAE.

**Generation of single-cell data.** Given a dataset $\{X, V, W\}$, where the vector $\mathbf{x} \in \mathbb{R}^M$ in the gene expression matrix $X$ represents gene expression vector of a cell, the vector $\mathbf{v} \in \mathbb{R}^K$ in the matrix $V$ represents cell type proportion, satisfying $\log(p(\mathbf{v}|\mathbf{x})) = \sum_k \log(p(\mathbf{v}_k|\mathbf{x}))$, where $\mathbf{v}$ is restricted by a loss function:

$$MAE = \sum_{\mathbf{v}} |\mathbf{v} - \hat{\mathbf{v}}| \tag{1}$$

Here $\hat{\mathbf{v}}$ is the predicted proportions of certain cell type.

The vector $\mathbf{w} \in \mathbb{R}^K$ in the matrix $W$ represents conditionally correlated generative factor. The factor $\mathbf{w}$ is obtained from the same class of cells through the β-VAE Encoder. For each class of cells, the average value after model training represents a class of cell-specific $\mathbf{w}$, and it is not restricted by adding a loss function. According to Higgins

et al.[26], we hypothesize that gene expression vectors $\mathbf{x}$ are generated by a probability model $p_{\boldsymbol{\theta}}(\mathbf{x}|\mathbf{v},\mathbf{w})$, where $\boldsymbol{\theta}$ represents the generative model parameters. The model learns the joint distribution of the data $\mathbf{x}$ and a set of latent variables $\mathbf{z}$ ($\mathbf{z} \in \mathbb{R}^M$, where $M \geq K$) for generating observed data $\mathbf{x}$, i.e., $p_{\boldsymbol{\theta}}(\mathbf{x}|\mathbf{z}) \approx p(\mathbf{x}|\mathbf{v},\mathbf{w})$, and approximates the true posterior distribution $p_{\boldsymbol{\theta}}(\mathbf{z}|\mathbf{x})$ with an approximate posterior distribution $q_{\boldsymbol{\phi}}(\mathbf{z}|\mathbf{x})$ that is easier to compute. Our goal is to ensure that the inferred latent variables $\mathbf{z}$ capture the generative factors $\mathbf{w}$ in a disentangled manner. A disentangled representation implies that individual latent unit is sensitive to variations in a single generative factor while being relatively invariant to variations in other factors. In a disentangled representation, knowledge of one factor can be generalized to new configurations of other factors. The conditionally correlated generative factors $\mathbf{w}$ can remain entangled in a separate subset of $\mathbf{z}$ and are not used to represent $\mathbf{v}$.

To achieve this, we minimize the KL divergence between the approximate posterior and the true posterior:

$$\mathcal{KL}(q_{\boldsymbol{\phi}}(\mathbf{z}|\mathbf{x})||p_{\boldsymbol{\theta}}(\mathbf{z}|\mathbf{x})) = -\sum_{\mathbf{z}} q_{\boldsymbol{\phi}}(\mathbf{z}|\mathbf{x}) \log(p_{\boldsymbol{\theta}}(\mathbf{x}|\mathbf{z})q_{\boldsymbol{\phi}}(\mathbf{z}|\mathbf{x})) + \log(p_{\boldsymbol{\theta}}(\mathbf{x}))$$

(2)

Here, $\mathcal{KL}(q_{\boldsymbol{\phi}}(\mathbf{z}|\mathbf{x})||p_{\boldsymbol{\theta}}(\mathbf{z}|\mathbf{x}))$ is the variational lower bound and can be written as:

$$\mathcal{L}(\boldsymbol{\theta},\phi,\mathbf{x}) = \sum_{\mathbf{z}} q_{\phi}(\mathbf{z}|\mathbf{x}) \log(p_{\boldsymbol{\theta}}(\mathbf{x}|\mathbf{z})) - \mathcal{KL}(q_{\phi}(\mathbf{z}|\mathbf{x})||p_{\boldsymbol{\theta}}(\mathbf{z}|\mathbf{x}))$$

(3)

We introduce a constraint to shape the inferred posterior $q_{\boldsymbol{\phi}}(\mathbf{z}|\mathbf{x})$ and match it with a prior $p_{\boldsymbol{\theta}}(\mathbf{z})$ that controls the capacity of the latent information bottleneck. We set the prior as an isotropic unit Gaussian, $p(\mathbf{z}) \sim \mathcal{N}(0,I)$. The constrained optimization problem can be written as:

$$\max_{\boldsymbol{\phi},\boldsymbol{\theta}} \mathbb{E}_{q_{\boldsymbol{\phi}}(\mathbf{z}|\mathbf{x})}\left[\log(p_{\boldsymbol{\theta}}(\mathbf{x}|\mathbf{z}))\right] \text{s.t.} \mathcal{KL}(q_{\boldsymbol{\phi}}(\mathbf{z}|\mathbf{x})||p(\mathbf{z})) < \epsilon$$

(4)

Here, $\epsilon$ is the strength of the applied constraint. With this optimization based on MLE, the latent variable $\mathbf{z}$ can reflect the character of the ground truth data with lower error. According to β-VAE model[31], we can rewrite the problem in Lagrangian form:

$$\mathcal{F}(\boldsymbol{\theta},\boldsymbol{\phi},\beta,\mathbf{x},\mathbf{z}) = \mathbb{E}_{q_{\boldsymbol{\phi}}(\mathbf{z}|\mathbf{x})}\left[\log(p_{\boldsymbol{\theta}}(\mathbf{x}|\mathbf{z}))\right] - \beta\left(\mathcal{KL}(q_{\boldsymbol{\phi}}(q_{\boldsymbol{\phi}}(\mathbf{z}|\mathbf{x})||p(\mathbf{z})) - \epsilon\right)$$

(5)

where $\beta$ is the regularization coefficient of the constraint, which limits the capacity of $\mathbf{z}$ and imposes an implicit pressure for independence in learning the posterior distribution due to the isotropic nature of the Gaussian prior $p_{\boldsymbol{\theta}}(\mathbf{z})$. In this model, different values of $\beta$ can alter the level of learning pressure imposed during training, encouraging the learning of different representations. We assume a disentangled representation of the conditional independent data generative factors $\mathbf{v}$ and therefore set $\beta > 1$ to apply a stronger constraint on the latent variable information bottleneck, exceeding the constraint of the original VAE. These constraints restrict the capacity of $\mathbf{z}$ and, combined with the pressure to maximize the log-likelihood of the training data $\mathbf{x}$, encourage the model to learn the most efficient representation of the data.

**Computation of single-cell neighborhood graph.** Here, we used the scanpy.pp.neighbors function from Scanpy to compute the cell neighborhood graph. For detailed mathematical description, please refer to the relevant papers and documentation of nearest neighbor descent in Scanpy and PyNNDescent[62].

**Community detection and generation of overlapping cell communities.** We performed community detection on the cell neighborhood graph using a Graph Neural Network (GNN) model to find overlapping

cell communities[33]. GNN can learn relationships between nodes and divide them into different communities based on their similarities. Specifically, we used GCN, which is one of the basic models in GNN, to generate an affinity matrix G, which represents the degree of association between cells. The computation is as follows:

$$G := GNN_{\boldsymbol{\theta}}(A,X)$$

(6)

Here, $A$ is the adjacency matrix of the cell neighborhood graph, and $\boldsymbol{X}$ represents cell type as the node feature. To ensure non-negativity of G, we applied element-wise ReLU non-linear activation function to the output layer. For detailed information about the GNN architecture,

$$G := GCN_{\boldsymbol{\theta}}(A,X) = ReLU\left(\hat{A} ReLU\left(A\hat{A}XW^{(1)}\right)W^{(2)}\right)$$

(7)

Here, $\hat{A} = \hat{D}^{-\frac{1}{2}}\widetilde{A}\hat{D}^{-\frac{1}{2}}$ is the normalized adjacency matrix, $\widetilde{A} = A + I_N$ is the adjacency matrix with self-loops, and $\hat{D}_{ii} = \sum_j \widetilde{A}_{ij}$ is the diagonal degree matrix of the adjacency matrix with self-loops. We considered other GNN architectures and deeper models but did not observe significant improvements. Two main differences between our model and the standard GCN are: (1) batch normalization applied after the first graph convolutional layer, and (2) L2 regularization applied to all weight matrices. We found that both modifications significantly improved the performance.

We measured the fit between the generated affinity matrix $F$ and the neighborhood graph using the negative log-likelihood function of the Bernoulli-Poisson model:

$$-\log p(A|F) = -\sum_{(u,v)\in E} \log(1 - \exp(-F_u F_v^T)) + \sum_{(u,v)\notin E} F_u F_v^T$$

(8)

Here, $E$ represents the set of edges in the graph. Since neighborhood graphs of single-cell data are typically sparse, the second term in the third sum contributes more to the loss. To balance these two terms, we adopted a standard technique known as balanced classification[18], and defined the loss function as follows:

$$L(F) = -\mathbb{E}_{(u,v)\sim P_E}\left[\log\left(1 - \exp\left(-F_u F_v^T\right)\right)\right] + \mathbb{E}_{(u,v)\sim P_N}\left[F_u F_v^T\right]$$

(9)

Here, $P_E$ and $P_N$ represent uniform distributions over edges and non-edges, respectively.

Instead of directly optimizing the affinity matrix $F$ as in traditional methods, we search for the optimal neural network parameters $\boldsymbol{\theta}^*$ to minimize the (balanced) negative log-likelihood function:

$$\boldsymbol{\theta}^* = \arg\min_{\boldsymbol{\theta}} L(GCN_{\boldsymbol{\theta}}(A,X))$$

(10)

Through these steps, the BulkTrajBlend model computes overlapping communities in single-cell data, which can be used to infer "omission" cells in the original single-cell data. It can help reveal cell type transitions and dynamics, and model and analyze cell developmental trajectories.

**Community trajectory inference.** Here, we inserted the overlapping communities of target cells into the original single-cell data and used PyVIA to infer pseudo-temporal trajectories of cell differentiation. For detailed inference methods, please refer to the mathematical description of PyVIA. Additionally, researchers can also use CellRank for community trajectory re-inference.

## CGAN and ACGAN model description

CGAN (Conditional Generative Adversarial Nets) is a GAN (Generative Adversarial Nets) based model that generates data by training the generator and discriminator with the data and corresponding labels. The training process can be split into 2 parts. In the first part, latent

variables $\mathbf{z} \in \mathbb{R}^{M}(M=100)$ are generated by standardized normal distribution and its generated class labels $\boldsymbol{l}_g$ are input into the generator to get the generated data. Here the generator can be summarized as a function $g_{\boldsymbol{\theta}}$, where $\boldsymbol{\theta}$ are the parameters of the MLP and there are 6 layers in that each layer is normalized. The the hidden dimensions are 128*256*512*1024 and the activation function is LeakyRelu. After getting the generated data $\boldsymbol{g}=g_{\boldsymbol{\theta}}(\mathbf{z},l_g)$, there will be a discriminator $d_{\boldsymbol{\phi}}$, where $\boldsymbol{\phi}$ are the parameters of the MLP and there are 4 layers in each layer the hidden dimension is 512, dropout rate is 0.4 and the activation function is LeakyRelu, judging whether $\boldsymbol{g}$ accords with its label $\boldsymbol{l}_g$. Therefore, in the second part, $d_{\boldsymbol{\phi}}$ will be trained by the real data $\boldsymbol{r}$ and its label $\boldsymbol{l}_r$ with Adam optimizer to improve the judgement level of $d_{\boldsymbol{\phi}}$. Then the loss of $g_{\boldsymbol{\theta}}$ judged by $d_{\boldsymbol{\phi}}$ will be employed to enhance the generation ability of $g_{\boldsymbol{\theta}}$ with the same optimizer. The loss functions for $g_{\boldsymbol{\theta}}$ and $d_{\boldsymbol{\phi}}$ are both MSEloss and the weights of the loss of the generative data and the real data are both 0.5.

In addition, ACGAN (Auxiliary Classifier GAN), which makes the generative data more authentic, keeps the same structure of the generator as the one in the CGAN, but it adds the classifier that offers the label of the input data on the output of the discriminator. In the training process, the loss function for the added classifier is CrossEntropy.

## Data pre-processing

All single-cell data used for BulkTrajBlend training underwent the same quality control steps: Cells with low sequencing counts (<1000) and a high mitochondrial fraction (>0.2) were excluded in further analysis. The filtered count matrix was normalized by dividing the counts of each cell by total molecule counts detected in that particular cell and logarithmised with Python library scanpy[63]. All Bulk RNA-seq were normalized using DEseq2 and "numpy.log1p" logarithmised using Python's Numpy[64] package. It is worth noting that both Bulk and single-cell data use raw counts during AE estimation of the cell fraction state space, whereas both Bulk and single-cell data use normalized and logarithmised data during training of β-VAE.

## Performance evaluation

To evaluated the generated and interpolation performance of our model, a comprehensive analysis was conducted, encompassing the examination of five critical dimensions:

(1) The count of interpolated cells, we counted the number of cells that were eventually used to interpolate into the raw single-cell profile.

(2) The correlation in marker gene expression between interpolated and authentic cells, we first use scanpy's "scanpy.tl.rank_genes_groups" function to calculate the marker genes for each type of cell subpopulation in the raw single-cell profile (taking the top 200 marker genes). Then, we use the Pearson coefficient to calculate the percentage of these 200 marker genes in the expression correlation between the generated single-cell profile and the raw single-cell profile.

(3) Marker gene similarity, we first used scanpy's "scanpy.tl.rank_genes_groups" function to calculate the marker genes for each type of cell subpopulation (taking the first 200 marker genes) in the raw single-cell profile versus the generated single-cell profile, respectively. Then, we treated marker genes as words and all the marker genes of each cell class as sentences, and used cosine similarity to calculate the similarity of marker genes of each cell subpopulation.

(4) Transition probabilities post-interpolation We first wrapped "omicverse.pp.scale" and "omicverse.pp.pca" in omicverse, "omicverse.utils.cal_paga", and computed the principal component PCA of the single-cell profile. We took the first 50 principal components and used the scanpy's "scanpy.pp.neighbour" to compute the neighborhood map of the single-cell profile. Immediately after that, we calculated the developmental trajectory of single-cell profile with pseudotime using pyVIA, and we calculated the state transfer confidence for each type of cell subpopulation by taking pseudotime as

the priority time with the neighborhood graph as the input of "omicverse.utils.cal_paga".

(5) The number of noise clusters, we used "scanpy.tl.leiden" in scanpy to perform unsupervised clustering on the generated single-cell profiles, with the resolution set to 1.0, and we identified the categories with less than 25 cells after clustering as noisy clusters and counted the number of noisy clusters as an assessment of the generation quality.

(6) Density assessment of pseudotime, after we obtained the pseudotime of single-cell profiles using pyVIA as the default parameters, specifically setting K to 15 in the neighborhood graph of the KNN and configuring use_rep to X_pca.We assessed the variance of the pseudotime of target interpolated cells as one of the metrics for the assessment of developmental trajectory reconstruction.

## Datasets

**Dentate Gyrus.** Single-cell RNA-seq: Data from Hochgerner et al.[65]. Dentate gyrus (DG) is part of the hippocampus involved in learning, episodic memory formation and spatial coding. The experiment from the developing DG comprises two time points (P12 and P35) measured using droplet-based scRNA-seq (10x Genomics Chromium). The dominating structure is the granule cell lineage, in which neuroblasts develop into granule cells. Simultaneously, the remaining population forms distinct cell types that are fully differentiated (e.g., Cajal-Retzius cells) or cell types that form a sub-lineage (e.g., GABA cells) (Accession ID GSE95753).

Bulk RNA-seq: Data from Cembrowski et al.[66]. Dentate gyrus (DG) is measured by RNA sequencing (RNA-seq) to produce a quantitative, whole genome atlas of gene expression for every excitatory neuronal class in the hippocampus; namely, granule cells and mossy cells of the dentate gyrus, and pyramidal cells of areas CA3, CA2, and CA1 (Accession ID GSE74985).

**Pancreatic endocrinogenesis.** Single-cell RNA-seq: Data from Bastidas-Ponce et al.[67]. Pancreatic epithelial and Ngn3-Venus fusion (NVF) cells during secondary transition with transcriptome profiles sampled from embryonic day 15.5. Endocrine cells are derived from endocrine progenitors located in the pancreatic epithelium. Endocrine commitment terminates in four major fates: glucagon- producing α-cells, insulin-producing β-cells, somatostatin-producing δ-cells and ghrelin-producing ε-cells (Accession ID GSE132188).

Bulk RNA-seq: Data from Bosch et al.[68]. RNA-sequencing was performed of pancreatic islets (islets of Langerhans) from mice on PLX5622 or control diet for 5.5 or 8.5 months (Accession ID GSE189434).

**Human bone marrow.** Single-cell RNA-seq: Data from Setty et al.[69]. The bone marrow is the primary site of new blood cell production or haematopoiesis. It is composed of hematopoietic cells, marrow adipose tissue, and supportive stromal cells. This dataset served to detect important landmarks of hematopoietic differentiation, to identify key transcription factors that drive lineage fate choice and to closely track when cells lose plasticity (https://data.humancellatlas.org/explore/projects/091cf39b-01bc-42e5-9437-f419a66c8a45).

Bulk RNA-seq: Data from Myers et al (2018). RNA-Seq of CD34+ Bone Marrow Progenitors from Healthy Donors (Accession ID GSE118944).

**Maturation of murine liver.** Single-cell RNA-seq: Data from Liang et al.[70]. A total of 52,834 single cell transcriptomes, collected from the newborn to adult livers, were analyzed. We observed dramatic changes in cellular compositions during liver postnatal development. We characterized the process of hepatocytes and sinusoidal endothelial cell zonation establishment at single cell resolution. We selected Pro-B, Large Pre-B, SmallPre-B, B, HPC, GMP, iNP, imNP, mNP, Basophil,

Monocyte, cDC1, cDC2, pDC, aDC, Kupffer, Proerythroblast, Erythroblast, erythrocyte (Annotation could be found in metadata of Data from Liang et al.) to performed HPC differentiation analysis (Accession ID GSE171993).

Bulk RNA-seq: Data from Renaud et al.[71]. We analyze gene expression patterns in the developing mouse liver over 12 distinct time points from late embryonic stage (2 days before birth) to maturity (60 days after birth). Three replicates per time point (Accession ID GSE58827).

### Construction of Simulated "omission" single-cell profile

To simulate the cell "omission" in single-cell sequencing, we conducted cell dropout experiments across diverse datasets. In the Pancreas dataset, we employed Leiden clustering and manually excluded specific clusters of Ngn3 high EP, resulting in a reduction of confidence in the transition from Ngn3 high EP to Pre-endocrine to 0. In the Dentategyrus dataset, we applied Leiden clustering and manually removed specific clusters of Granule Immature, leading to a confidence reduction in the transition from Granule Immature to Granule Mature to 0. Furthermore, in the BoneMarrow dataset, we randomly eliminated 80% of the cells from HSC-2, causing a confidence drop in the transition from HSC-2 to Monocyte-2 to 0.

To employ BulkTrajBlend for generating "omission" cells across various datasets, we generated single-cell data from the bulk RNA-seq data using BulkTrajBlend and filtered out noisy cells using the size of the Leiden as a constraint. In configuring the model for different datasets, we set the hyperparameter "cell_target_num" to be 1.5 times, 1 time, and 6 times the number of dropped-out cell types, aligning with Pancreas, Dentategyrus, and BoneMarrow, respectively. Subsequently, BulkTrajBlend calculated the overlapping cell types in the generated single-cell data, and we annotated the overlapping cell communities. Specifically, we selected the single-cell data in which dropped-out cell types were associated with adjacent cell types.

### Methods of OmicVerse integration

We unified the downstream analyses of Bulk RNA-seq, single cell RNA-seq in OmicVerse. Since the downstream analyses are independent of the parameter evaluation of BulkTrajBlend and the analysis modules of each part are independent of each other, we have placed the datasets and methods used in each part in Supplementary, an index of which is provided here.

(1) Bulk RNA-seq: All datasets selected, parameter setting, and methods could be found in Supplementary Note 4.
(2) scRNA-seq: All datasets selected, parameter setting, and methods could be found in Supplementary Note 5.
(3) Multi-omics: All datasets selected, parameter setting, and methods could be found in Supplementary Note 6.

### Reporting summary

Further information on research design is available in the Nature Portfolio Reporting Summary linked to this article.

## Data availability

The Dentate Gyrus data used in this study have been deposited in the Gene Expression Omnibus (GEO) database under accession code GSE95753 and GSE74985, Data related to pancreatic endocrinogenesis are accessible via accession codes GSE132188 and GSE189434, the maturation of murine liver data can be found under accession code GSE171993 and GSE58827, Human bone marrow data are available in the Human Cell Atlas (HCA) database at https://data.humancellatlas.org/explore/projects/091cf39b-01bc-42e5-9437-f419a66c8a45 and in the GEO database under accession code GSE118944. The Alzheimer's Disease snRNA-seq and snATAC-seq used in this study are available from GSE174367. The colorectal cancer scRNA-seq data is available

from GSE178318. All processed data in this manuscript are available at https://github.com/Starlitnightly/omicverse-reproducibility.

## Code availability

The code to reproduce the experiments of this manuscript is available at https://github.com/Starlitnightly/omicverse-reproducibility. The OmicVerse package can be found on GitHub at https://github.com/Starlitnightly/omicverse Documentation and tutorials can be found at https://omicverse.readthedocs.io.

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

## Acknowledgements

This work was supported by the grants from the National Natural Science Foundation of China (32300682 to C.X.), the National Key Research & Developmental Program of China (92249303 to Y.X.), the Fundamental Research Funds for the Central Universities (FRF-TP-22-007A1 to C.X.), the Student Research Training Program (SRTP) of University of Science and Technology Beijing (202010008107 to Z.Z.). We thank Professor Ge Gao of Peking University for his guidance on the OmicVerse open-source copyright in the summer of 2021. We are grateful for the experience of studying epigenomics in the Xie Lab at Tsinghua University, and for Xiaotong Wu's guidance in enabling OmicVerse's multi-omics analyses to be successfully designed. We thank all the Github users who contributed code and issue to OmicVerse over the years. We would like to thank the following WeChat Official Accounts for promoting OmicVerse: pythonic biologists, biotrainee. Pythonic biologist and Biotrainee's article inspired some of the charting in the OmicVerse.

## Author contributions

Z.Z., Y.M. and L.H. contributed equally to this work. Z.Z. was responsible for designing the OmicVerse application programming interface and designing the whole BulkTrajBlend framework. Y.M. played a key role in designing and implementing the overlap cell community of BulkTrajBlend, while Z.Z. was responsible for implementing and conducting testing of the single-cell RNA-seq pipeline. L.H. conducted simulated single-cell profile tests for BulkTrajBlend. The multi-omics module of pyMOFA was implemented and tested by L.H., Y.W. and Z.Z. P.L. handled the implementation and testing of the meta cells analysis in SEACells. B.T. was responsible for writing the methods for CGAN and ACGAN, as well as reviewing the methods of BulkTrajBlend. H.D., Y.X. and Z.Z. jointly conceived, implemented, and tested the bulk RNA-seq pipeline. C.X. and Y.X. provided the conceptualization of false overlap rate for evaluation of BulkTrajBlend. Y.X., H.D., C.X. and Z.Z. provided supervision and contributed to the conceptualization of the OmicVerse platform. The manuscript was collaboratively written by Z.Z., Y.M., L.H., H.D. and Y.X.

## Competing interests

The authors declare no competing interests.
