## [Peer Review File · Nature Communications]

OmicVerse: A Framework for Bridging and Deepening Insights Across Bulk and Single-Cell SequencingREVIEWER COMMENTS

Reviewer #1 (Remarks to the Author):

In the manuscript "OmicVerse: A single pipeline for exploring the entire transcriptome universe", the authors present their Python library OmicVerse for processing and analyzing bulk and single-cell RNA-seq data. The paper is structured into two parts: (1) a description of a new algorithm called BulkTrajBlend for inferring "interrupted" cells in single-cell RNA-seq data. And (2), an overview of the other methods provided by OmicVerse for analyzing bulk and single-cell RNA-seq data.

The manuscript is well structured, and the Python package is comprehensively documented online, which will help users. However, the main problem of the manuscript is a lack of validation and comparison with existing literature, which makes it hard to assess the contribution of the work.

Major Concerns

The goal of the BulkTrajBlend algorithm is to identify "interrupted" cells. This is a new concept, yet the manuscript offers no clear definition of what is meant by the term "interrupted".

BulkTrajBlend tries to infer the expression pattern of cells in developmental trajectories between two states which have not been measured. To achieve this, the authors posit that these cells would be measured in bulk RNA-seq (Suppl. lines 78-81). The authors should explain why they think this is the case and how the deconvolution method can infer these cell states if they are missing from the reference single-cell data.

The authors evaluate the performance of BulkTrajBlend using mouse dentate gyrus single and bulk RNA-seq data. As evidence for the success of their algorithm, they use the fact that after augmenting the raw single-cell data with the generated single-cells from the bulk data, the OPC cells are located close to the nIPC cells (Fig. 1i). To make the evaluation more convincing, the authors should quantify the effect as UMAP visualizations are notoriously unreliable when it comes to the global position of clusters. Furthermore, it is unclear if the choice to "selected the single-cell data in which 81 OPCs were associated with nIPC" (line 80f) biased the result.

BulkTrajBlend is built around the beta variational autoencoder, originally developed by Higgins et al. (2017). The authors fail to cite this paper, despite its importance for this manuscript. This is particularly unfortunate as their method section (Suppl. line 93ff) closely follows the structure of the text and mathematical notation in Higgins et al..

In the second half, the authors describe the additional functionality of OmicVerse for analyzing single-cell and bulk RNA-seq data. OmicVerse wraps functions from many existing Python packages and provides a uniform interface. However, the lack of a comparison with scanpy --a well-established Python package for single-cell data analysis-- makes it difficult to judge the added benefit of the package.

Minor issues

- * Subpanels A and B of Fig. 1 are not mentioned in the text.
- * The order of the subpanels does not match the order in which they are discussed in the text (that is, Fig 1i is referred to first (line 68) before referring to panels c and d (line 76)).
- * The deconvolution step of the bulk2single function produces very different cell type proportions than observed in the reference data. The authors explain that this is because the single-cell sequencing did not pick up neurons (Suppl. line 247). They should invest more effort to demonstrate that this is indeed a desired outcome of the algorithm and not an artifact.
- * In Fig. 2b and the online documentation, the authors use the term "different expression analysis"; this should be "differential expression analysis".
- * Fig. 1e lacks axis labels.
- * Fig. 1j lacks a legend for the color scheme.
- * The authors should provide more details on the benefits of choosing a graph neural network (step 3) over other clustering algorithms.
- * In Suppl. line 84f, the authors state that "the UMAP method is employed to compute the neighborhood graph" which is odd. UMAP is a visualization technique for high-dimensional data. In

Suppl. line 144, the authors correctly describe using the `scanpy.pp.neighbors` function.
* In Suppl. line 101, the authors refer to K ; however, this letter is not introduced early.
* In Suppl. line 417, there appear to be some font display problems as the reference is rendered as Chinese characters.
* In line 73, the authors write "deconvoluted"; this should be "deconvolved".
* In line 125, the authors write in the formula $\mathbb{E} * q * \phi \dots$; the notation is non-standard and needs further explanation.

Reviewer #2 (Remarks to the Author):

The manuscript introduces OmicVerse, a comprehensive bioinformatics tool designed for single cell and bulk-seq analyses. Central to this tool is the proposed method, BulkTrajBlend, which employs beta-VAE to generate single cell data from bulk-seq and subsequently interpolates the data via GNN and community analysis. This conceptual approach holds promise for a wide range of research endeavors. However, the manuscript and the methodology, in their present form, have noteworthy limitations that warrant attention.

Manuscript Structure and Presentation:

The current presentation of the manuscript more closely resembles an informal letter rather than a structured research article. Given its submission as an article, a comprehensive restructuring is necessary to meet the expectations of a peer-reviewed scientific publication.

While OmicVerse is extensively documented and has the potential to serve as a holistic bioinformatics toolbox, the manuscript appears to lack a clear research focus. It would be beneficial for the authors to concentrate their discussion on BulkTrajBlend, elucidating how this method seamlessly integrates into the OmicVerse framework to provide value to the scientific community.

Validation of BulkTrajBlend:

The foundational idea of BulkTrajBlend—to generate single cell data from bulk-seq using beta-VAE and interpolate single cell communities with GNN—is intriguing. Nevertheless, there appears to be an absence of rigorous validation with well-established ground-truth datasets. While the authors allude to the nIPC to CPN dataset, a more thorough exploration would enhance the manuscript's credibility.

To further demonstrate the efficacy of BulkTrajBlend, the authors could consider applying it to existing datasets that have both bulk and single cell data. By simulating varying conditions of cell dropout, the authors could showcase BulkTrajBlend's ability to accurately discern cell transitions between different states.

Benchmarking and Parameter Sensitivity:

The authors should rigorously benchmark BulkTrajBlend against other approaches employed in single cell studies, specifically those utilizing either the Variational Autoencoder (VAE) or Generative Adversarial Network (GAN) frameworks as well as other bulk-deconvolution approaches.

Given the potential vastness of the parameter space associated with this method, a comprehensive benchmarking study is essential. The authors should test a diverse set of parameters to determine how variations might significantly influence the results.

Addressing Potential False Positives:

Another concern for BulkTrajBlend is to produce false positives. For instance, in mature tissues, certain cell states might lack transitional phases. How does BulkTrajBlend handle such scenarios? Does it erroneously predict "interrupted" cells? A quantitative exploration of this aspect is imperative for the tool's broader acceptance and trustworthiness.

In conclusion, while the manuscript introduces a promising method in the realm of bioinformatics, several critical improvements are needed to elevate its scientific rigor and impact.

Point-to-point replies to the reviewers' comments:

In the following, we present our response to the reviewers' comments. We give *comments (blue)*, *point-by-point answers (green)* to the questions, and in parts copy parts of the text or specific panels (black), which directly correspond to comments or reference to them.

We express our gratitude to the esteemed reviewers for their valuable comments, recommendations, and constructive feedback.:

- 1) **Reformatting the manuscript.** We have reformatted the manuscript from a letter to an article, aligning it with the requirements for publication. Additionally, we have integrated Supplementary Notes 4 (OmicVerse provides a multi-type pipeline for single-cell RNA-seq analysis), 5 (OmicVerse provides a comprehensive analysis platform for bulk RNA-seq data), and 6 (OmicVerse performs multi-omics factor analysis with MOFA and GLUE) into the main text. Fig.1 from the original main text is now presented in Supplementary Note 2, serving as an illustrative example of BulkTrajBlend.
- 2) **Clarify of simulation and hyperparameter selected results.** To enhance clarity, we have redefined the algorithmic model of BulkTrajBlend (Fig.1) inspired by the TAPE algorithm. Specifically, in the bulk2single component of BulkTrajBlend, we introduced the cell state space to establish a more precise connection between Bulk RNA-seq and scRNA-seq. Moreover, for benchmarking, we incorporated comparisons involving different generative algorithm kernels, such as ACGAN and CGAN. Additionally, six quantitative metrics have been introduced to assess the generative power of BulkTrajBlend on both real datasets and simulated "interruption" datasets. We meticulously examined the impact of various hyperparameter choices on model generation.

Reviewer#1:

In the manuscript "OmicVerse: A single pipeline for exploring the entire transcriptome universe", the authors present their Python library OmicVerse for processing and analyzing bulk and single-cell RNA-seq data. The paper is structured into two parts: (1) a description of a new algorithm called BulkTrajBlend for inferring "interrupted" cells in single-cell RNA-seq data. And (2), an overview of the other methods provided by OmicVerse for analyzing bulk and single-cell RNA-seq data.

The manuscript is well structured, and the Python package is comprehensively documented online, which will help users. However, the main problem of the manuscript is a lack of validation and comparison with existing literature, which makes it hard to assess the contribution of the work.

Major Concerns

1. *The goal of the BulkTrajBlend algorithm is to identify "interrupted" cells. This is a new concept, yet the manuscript offers no clear definition of what is meant by the term "interrupted".*

Response: We appreciate the reviewer's insightful comment. While BulkTrajBlend proficiently interpolates "interrupted" cells, it's important to clarify that the algorithm doesn't autonomously identify these cells; rather, such identification is a user-driven process. In the initial manuscript, we elaborated on the notion that "interrupted" cells in single-cell sequencing

stem from inherent limitations. Recognizing the lack of clarity in our original explanation, we have incorporated the following clarification in the Main section during the introduction of BulkTrajBlend:

Integrating single-cell and bulk sequencing results can be intricate, producing complex, multi-layered data sets that challenge the extraction of meaningful biological insights. A recognized impediment in single-cell sequencing is the 'interruption' -- the omission of certain cell types due to technological constraints on the sequencing platform and interruption the trajectory of cell differentiation, such as the enzymatic lysis-related loss of podocytes and intercalated cells¹, the differentiation from HPC to podocytes was interrupted, and the filtering-induced absence of neutrophils, cardiomyocytes, neuronal cells, and megakaryocytes and the differentiation from nIPC to neurons was interrupted²⁻⁴. The BD Rhapsody™ single-cell platform overcomes granulocyte loss by accommodating their natural sedimentation⁵. Conversely, bulk RNA-seq of whole tissues intrinsically includes these 'interrupted' cells. Current algorithm for isolating 'interrupted' cells from bulk RNA-seq are non-existent, revealing a gap in the tools available for reconciling bulk RNA-seq and scRNA-seq data.

- BulkTrajBlend tries to infer the expression pattern of cells in developmental trajectories between two states which have not been measured. To achieve this, the authors posit that these cells would be measured in bulk RNA-seq (Suppl. lines 78-81). The authors should explain why they think this is the case and how the deconvolution method can infer these cell states if they are missing from the reference single-cell data.*

Response: We appreciate the thoughtful inquiry from the reviewer. Addressing the question of inferring expression patterns in developmental trajectories involving unmeasured cell states, it is crucial to note that complete reconstruction is not feasible if a particular class of cells is entirely absent in the reference scRNA-seq. However, for partially present cells, exemplified by Basophil, as elucidated in the revised manuscript, their partial representation in reference scRNA-seq poses a challenge. Particularly, rare cell types like Basophil, with limited cells in the dataset by sequencing, result in the absence of intermediate state cells differentiating from haematopoietic stem cells to Basophil. To tackle this, BulkTrajBlend employs an innovative strategy. The cell state space serves as the output of the encoder in the autoencoder (AE). We utilize reference scRNA-seq to create a simulated Bulk RNA-seq for self-encoder training. After optimizing the model, we employ real Bulk RNA-seq as the input for the encoder, training it to obtain the state space of the input real Bulk RNA-seq. This process allows us to deduce the state space of "Interruption" from Bulk RNA-seq. Detailed methodologies of BulkTrajBlend are thoroughly explicated in the Methods section (pages from 20 in manuscript).

- The authors evaluate the performance of BulkTrajBlend using mouse dentate gyrus single and bulk RNA-seq data. As evidence for the success of their algorithm, they use the fact that after augmenting the raw single-cell data with the generated single-cells from the bulk data, the OPC cells are located close to the nIPC cells (Fig. 1i). To make the evaluation more convincing, the authors should quantify the effect as UMAP visualizations are notoriously unreliable when it comes to the global position of clusters. Furthermore, it is unclear if the choice to "selected the*

single-cell data in which 81 OPCs were associated with nIPC" (line 80f) biased the result.

Response: We thank this reviewer for raising a pertinent concern regarding the reliability of UMAP visualizations in depicting the global position of clusters. In this revised version, we have incorporated two quantitative metrics to robustly evaluate the impact of "interruption" post-cell interpolation. Firstly, we introduce the probability of cell state transition based on PAGA, and secondly, we employ pseudotime density after interpolation. Detailed results and parameters pertaining to these metrics are comprehensively outlined in our Results section, specifically in "Impact of Varied Hyperparameters on Interpolation Performance in BulkTrajBlend" and "Proficient Reconstruction of Cell Developmental Trajectories in Simulated "Interruptions" Single-Cell Profiles" expounded upon in the Methods section. Additionally, addressing the reviewer's query about the selection of single-cell data wherein 81 OPCs were associated with nIPC, we meticulously evaluated various cell scales. Our findings, presented in the Results section under "Impact of Varied Hyperparameters on Interpolation Performance in BulkTrajBlend" (pages 6 in manuscript), demonstrate that employing 2x or 4x "Interruption" cell scales as inputs yielded optimal results for the model.

- 4. BulkTrajBlend is built around the beta variational autoencoder, originally developed by Higgins et al. (2017). The authors fail to cite this paper, despite its importance for this manuscript. This is particularly unfortunate as their method section (Suppl. line 93ff) closely follows the structure of the text and mathematical notation in Higgins et al.*

Response: We express appreciation to the reviewer for pointing out the omission of proper citation to the work of Higgins et al. (2017), a crucial foundation for BulkTrajBlend. We acknowledge that this oversight was unintentional, and to rectify it, we have promptly added citations to the relevant literature. The Result section, specifically under "Design concept of BulkTrajBlend and Benchmarking," now appropriately references the significance of the beta-variational autoencoder (β -VAE) developed by Higgins et al. (2017).

The conceptualization of BulkTrajBlend draws upon prior research, positing that Bulk RNA-seq data is a composite of scRNA-seq data through a nonlinear superposition mechanism^{6,7}. At the heart of this notion is the implementation of the beta-variational autoencoder (β -VAE), which serves as a powerful tool to approximate Bulk RNA-seq data to scRNA-seq representation^{8,9}. The incorporation of β -VAE facilitates the construction of an encoder and decoder from single-cell data, traditionally characterized by unconstrained attributes.

- 5. In the second half, the authors describe the additional functionality of OmicVerse for analyzing single-cell and bulk RNA-seq data. OmicVerse wraps functions from many existing Python packages and provides a uniform interface. However, the lack of a comparison with scanpy --a well-established Python package for single-cell data analysis-- makes it difficult to judge the added benefit of the package.*

Response: We sincerely appreciate the reviewer's insightful observation. SCANPY, initially designed for fundamental scRNA-seq analyses, lacks certain functionalities covered by

OmicVerse. The latter was conceived to create an extensive RNA-seq analysis framework, encompassing analyses in both Bulk RNA-seq and scRNA-seq. The distinctive contributions of OmicVerse can be outlined as follows:

- 1) Downstream analysis of Bulk RNA-seq in Python: OmicVerse pioneers comprehensive downstream analyses for Bulk RNA-seq in Python, re-implementing R-exclusive packages within the Python environment. This includes data integration, batch effect processing, differentially expressed gene analysis, gene set enrichment analysis, gene co-expression module identification, gene/protein interaction network identification, and basic processing and analysis of TCGA databases. The package eliminates the need for rpy2 in Python, providing a streamlined workflow for common Bulk RNA-seq analyses.
- 2) Downstream analysis of scRNA-seq in Python: Leveraging SCANPY as a foundation, OmicVerse optimizes quality control principles by expanding criteria and employing robust outlier identification methods. The introduction of pearson residuals enhances the identification of normalized and highly variable genes, aligning with single-cell optimal practices. OmicVerse introduces GPU-accelerated UMAP for efficient dimensionality reduction and visualization of large-scale datasets, offering advantages in handling single-cell datasets of substantial sizes. Integration of various toolkits and models, including Harmony, scanronam, pyCombat, pySCSA, MetaTiME, pyVIA, cellFateGenie, CellPhoneDB, and scDrug, ensures a unified interface, minimizing version conflicts and dependency issues during installation.
- 3) Multi-omics analysis in Python: OmicVerse extends its capabilities to multi-omics analysis, particularly focusing on scATAC-seq. By integrating GLUE's multimodal integration and the MOFA algorithm, OmicVerse provides cross-modal analysis functionalities in addition to scRNA-seq analysis.

We put a comparative analysis of the different frameworks in **Supplement Note 7 and show in Response to Review #1 Fig.1 also.**

The accompanying overview Extended Data figure 9 presents the various stages of the RNA-seq analysis and highlights differences between popular frameworks used for this purpose. omicverse has the same well-established ecosystem for scRNA-seq analysis as seurat, which makes up for the analysis of advanced functions in scanpy, such as automatic annotation of cell types, gene perturbation analysis, cellular interaction analysis, and drug response prediction. In addition, unlike the seurat and scanpy ecosystems, omicverse has a unique Bulk RNA-seq analysis system.

				Data reading 	Start from count matrices	✓	✓	✓
Quality control 	QC metrics	✓	✓	✓
	Doublet removal	✓	✓	✓
	Highly Variable Gene Calculating	seurat_v3 pearsonr	seurat	seurat_v3 pearsonr
Dimensionality reduction 	principal component analysis	✓	✓	✓
	Latent Dirichlet Allocation (LDA)	✗	✓	✓
	Visualization	UMAP/TSNE	UMAP/TSNE	UMAP/TSNE/MDE
Annotation 	Clustering	Leiden/Louvain	Leiden/Louvain	Leiden/Louvain/ GaussianMixture
	Find Marker	T test/Wilcoxon	Wilcoxon/logistic regression/ ROC/DESeq2	T-test/Wilcoxon /DESeq2/COSG
	Celltype automatically identity	✗	✗	pySCSA/MetaTIME/ Celltypist
Data integration 	Batch correction	Harmony/pyCombat/ FastMNN	CCA/RPCA/ Harmony/FastMNN	Harmony/pyCombat/scan orama/SIMBA/scVI/Mira
	Integration with scATAC-seq	✗	✓	✓
	Metacells/pseudobulk	✗	✓	✓
	Interpolation from Bulk RNA-seq	✗	✗	✓
	Deconvolution Bulk RNA-seq	✗	✗	✓
Trajectory inference 	Diffusion map	✓	✓	✓
	Pseudotime Calculated	PAGA graph	monocle3	pyVIA/Palantir
	Gene perturbation analysis	✗	✗	✓(using celloracle)
Cell structure 	Cell interaction	✗	✗	✓(using CellPhoneDB)
	Geneset score	✓	✓	✓
	Drug Response predicted	✗	✗	✓(using scDrug)

Response to Review #1 Fig 1 | Overview of the RNA-seq analysis ecosystem. Python: scanpy and omicverse, R: Seurat.

Minor issues

1. Subpanels A and B of Fig. 1 are not mentioned in the text.

Response: We sincerely appreciate the reviewer's diligence in identifying the omission of mentioning Subpanels A and B in Fig. 1. Recognizing this oversight, we have extensively revised the Supplement to ensure accurate referencing of the original Fig. 1 in the revised version.

2. The order of the subpanels does not match the order in which they are discussed in the text (that is, Fig 1i is referred to first (line 68) before referring to panels c and d (line 76)).

Response: We sincerely appreciate the reviewer's keen observation regarding the mismatch in the order of subpanels in Fig. 1. To rectify this discrepancy, we have diligently revised the order of the subpanels in the revised supplementary figures.

- The deconvolution step of the bulk2single function produces very different cell type proportions than observed in the reference data. The authors explain that this is because the single-cell sequencing did not pick up neurons (Suppl. line 247). They should invest more effort to demonstrate that this is indeed a desired outcome of the algorithm and not an artifact.*

Response: We express our gratitude to the reviewer for highlighting the concern regarding the disparity in cell type proportions resulting from the deconvolution step in the bulk2single function. Acknowledging this issue, we have implemented revisions to the bulk2single component in BulkTrajBlend, drawing inspiration from the approach employed in TAPE. Specifically, we have incorporated a methodology that involves simulating scRNA-seq with Bulk RNA-seq to ensure accurate estimation of cell proportions. This adjustment has been validated by comparing the simulated scRNA-seq with Bulk RNA-seq against real scRNA-seq data derived from Bulk RNA-seq, affirming the algorithm's enhanced ability to accurately estimate cell proportions. Based on BulkTrajBlend's cell ratio prediction, we found that granule cells were predominantly immature granule and mature granule in Bulk RNA-seq, and mossy cells were predominantly mossy in Bulk RNA-seq. whereas in CA1, CA2, CA3, the ratio of granule to mossy was close (Response to Review #1 Fig.2), which is consistent with the actual distribution of neurons in the mouse hippocampal region. Therefore, in the new BulkTrajBlend will effectively identify the proportion of different types of cells.

Response to Review #1 Fig 2 Neuronal fraction predicted in the hippocampus. The horizontal coordinate represents the sample name for Bulk RNA-seq and the vertical coordinate represents the percentage of cells predicted.

4. *In Fig. 2b and the online documentation, the authors use the term "different expression analysis"; this should be "differential expression analysis".*

Response: We appreciate the reviewer's attention to detail regarding the terminology used in Fig. 2b and the online documentation. In response, we have diligently corrected the term from "different expression analysis" to the accurate and standard "differential expression analysis" in all relevant sections.

5. *Fig. 1e lacks axis labels.*

Response: We acknowledge the reviewer's observation regarding the absence of axis labels in Fig. 1e. To address this, we have incorporated the necessary axis labels in Extended Data Fig. 1c, corresponding to the raw data in Fig. 1e.

6. *Fig. 1j lacks a legend for the color scheme.*

Response: We appreciate the reviewer's feedback, and in response, we have introduced a legend for the color scheme of pseudotime in Extended Data Fig. 1i, corresponding to the raw data presented in Fig. 1j. This addition enhances the interpretability and completeness of the figure.

7. *The authors should provide more details on the benefits of choosing a graph neural network (step 3) over other clustering algorithms.*

Response: We extend our gratitude to the reviewer for emphasizing the importance of providing more details on the benefits of choosing a graph neural network in Step 3. In response to this valuable suggestion, we have incorporated the following information into the Results section:

Most of the existing community discovery (cell clustering) algorithms are non-overlapping¹⁰, real communities are overlapping¹¹, and the GNN-based Neural Overlapping Community Detection (NOCD) algorithm achieves the best level in the existing baseline¹². Using NOCD enabling the identification of overlapping cell communities. This insight is integral to the subsequent task of recovering and reconstructing cell differentiation trajectories within the scRNA-seq data (Fig.1b).

8. *In Suppl. line 84f, the authors state that "the UMAP method is employed to compute the neighborhood graph" which is odd. UMAP is a visualization technique for high-dimensional data. In Suppl. line 144, the authors correctly describe using the `scanpy.pp.neighbors` function.*

Response: We appreciate the reviewer's astute observation and clarification regarding the use of UMAP in Suppl. Line 84f. To provide additional context, UMAP serves a dual purpose beyond its role as a visualization technique for high-dimensional data. In the

scanpy.pp.neighbors function, the method parameter defaults to umap, with the optional parameter gauss. The neighbor search efficiency in this context heavily relies on UMAP, which also offers a method for estimating connectivities of data points – the connectivity of the manifold (method=='umap').
(<https://scanpy.readthedocs.io/en/stable/generated/scanpy.pp.neighbors.html>).

9. *In Suppl. line 101, the authors refer to K ; however, this letter is not introduced early.*

Response: We appreciate the reviewer's identification of the missing introduction of the symbol K in Suppl. Line 101. To address this, we have introduced K in the definition of vector v , where K represents the dimension of the vector v that signifies the cell type proportion. The formula has been amended to reflect this inclusion:

$$v \in R^K$$

10. *In Suppl. line 417, there appear to be some font display problems as the reference is rendered as Chinese characters.*

Response: We acknowledge the font display problem identified by the reviewer in Suppl. Line 417. We have rectified the display error in the revised version.

11. *In line 73, the authors write "deconvoluted"; this should be "deconvolved".*

Response: We appreciate the reviewer's keen observation. The word "deconvoluted" in Line 73 has been corrected to "deconvolved" in the revised version.

12. *In line 125, the authors write in the formula $\mathbb{E} * q * \phi \dots$; the notation is non-standard and needs further explanation.*

Response: We have modified the formula and added explanation as shown below:

$$\max_{\phi, \theta} \mathbb{E}_{q_{\phi}(z|x)} [\log(p_{\theta}(x|z))] \quad \text{s. t.} \quad \mathcal{KL}(q_{\phi}(z|x) || p_{\theta}(z)) < \epsilon$$

Here, ϵ is the strength of the applied constraint. With this optimization based on MLE, the latent variable z can reflect the character of the ground truth data with lower error.

Reviewer#2:

The manuscript introduces OmicVerse, an comprehensive bioinformatics tool designed for single cell and bulk-seq analyses. Central to this tool is the proposed method, BulkTrajBlend, which employs beta-VAE to generate single cell data from bulk-seq and subsequently interpolates the data via GNN and community analysis. This conceptual approach holds promise for a wide range of research endeavors. However, the manuscript and the methodology, in their present form, have noteworthy limitations that warrant attention.

Manuscript Structure and Presentation:

1. *The current presentation of the manuscript more closely resembles an informal letter rather than a structured research article. Given its submission as an article, a comprehensive restructuring is necessary to meet the expectations of a peer-reviewed scientific publication.*

Response: We express our gratitude to the reviewer for highlighting the need for a more structured presentation in line with the expectations of a peer-reviewed scientific publication. Recognizing this, the manuscript has undergone a significant revision in the revised version. The logic of our study has been rearranged to adopt a more traditional research article format. Specifically, the Results section in the main text has been subdivided into six sections to enhance clarity and coherence: (1) Design concept of BulkTrajBlend and Benchmarking, (2) Impact of Varied Hyperparameters on Interpolation Performance in BulkTrajBlend, (3) Proficient Reconstruction of Cell Developmental Trajectories in Simulated "interruptions" Single-Cell Profiles, (4) OmicVerse provides a comprehensive analysis platform for bulk RNA-seq data, (5) OmicVerse provides a versatile multifaceted framework for scRNA-Seq Analysis, (6) OmicVerse performed multi-omics analysis with MOFA and GLUE.

2. *While OmicVerse is extensively documented and has the potential to serve as a holistic bioinformatics toolbox, the manuscript appears to lack a clear research focus. It would be beneficial for the authors to concentrate their discussion on BulkTrajBlend, elucidating how this method seamlessly integrates into the OmicVerse framework to provide value to the scientific community.*

Response: We express our appreciation to the reviewer for emphasizing the need for a clearer research focus, particularly concentrating on BulkTrajBlend within the OmicVerse framework. In response to this valuable feedback, we have refocused the manuscript to underscore the significance of BulkTrajBlend as the central component of the bridge between Bulk RNA-seq and scRNA-seq within the OmicVerse tool. The revised manuscript underscores the comprehensive evaluation of BulkTrajBlend's performance, encompassing benchmarking, hyperparameter selection, and interpolation on real and simulated datasets. This thorough assessment solidifies the role of BulkTrajBlend as a pivotal bridge in the OmicVerse framework. The paper now dedicates specific sections to elaborate on the analysis paradigms of the Bulk RNA-seq module and the scRNA-seq module, emphasizing their connection to the two sides of the bridge. Additionally, the multimodal integration of ATAC-seq, closely linked to RNA transcriptional regulation, is discussed within the context of OmicVerse.

Validation of BulkTrajBlend:

3. *The foundational idea of BulkTrajBlend—to generate single cell data from bulk-seq using beta-VAE and interpolate single cell communities with GNN—is intriguing. Nevertheless, there appears to be an absence of rigorous validation with well-established ground-truth datasets. While the authors allude to the nIPC to CPN dataset, a more thorough exploration would enhance the manuscript's credibility.*

Response: We appreciate the reviewer's insightful observation regarding the need for enhanced validation of BulkTrajBlend. In response to this, the revised manuscript introduces a new real

dataset and three simulated "interruption" dataset to provide a more comprehensive evaluation of BulkTrajBlend's performance. Reviewer could find it in the section of “(1) Design concept of BulkTrajBlend and Benchmarking, (2) Impact of Varied Hyperparameters on Interpolation Performance in BulkTrajBlend, (3) Proficient Reconstruction of Cell Developmental Trajectories in Simulated "interruptions" Single-Cell Profiles,”. For the real dataset, we incorporated a single-cell dataset focusing on liver hematopoietic development, in addition to the previously mentioned mouse dentate gyrus neurons. This expanded validation process aims to further establish the robustness of BulkTrajBlend across diverse datasets. To ensure a rigorous assessment, we have designed a set of six metrics, offering a quantitative evaluation of BulkTrajBlend's performance and resilience in various scenarios.

(1) The count of interpolated cells, we counted the number of cells that were eventually used to interpolate into the raw single-cell profile.

(2) The correlation in marker gene expression between interpolated and authentic cells, we first use scanpy's `scanpy.tl.rank_genes_groups` function to calculate the marker genes for each type of cell subpopulation in the raw single-cell profile (taking the top 200 marker genes), and then we use the Pearson coefficient to calculate the percentage of these 200 marker genes in the expression correlation between the generated single-cell profile and the raw single-cell profile.

(3) marker gene similarity, we first used scanpy's `scanpy.tl.rank_genes_groups` function to calculate the marker genes for each type of cell subpopulation (taking the first 200 marker genes) in the raw single-cell profile versus the generated single-cell profile, respectively, and then we will marker genes as words and all the marker genes of each cell class as sentences, and used cosine similarity to calculate the similarity of marker genes of each cell subpopulation.

(4) Transition probabilities post-interpolation We first wrapped `omicverse.pp.scale` and `omicverse.pp.pca` in omicverse, `omicverse.utils.cal_paga`, and computed the principal component PCA of the single-cell profile, we took the first 50 principal components and used the scanpy's `scanpy.pp.neighbour` to compute the neighbourhood map of the single-cell profile. Immediately after that we calculated the developmental trajectory of single-cell profile with pseudotime using pyVIA, and we calculated the state transfer confidence for each type of cell subpopulation by taking pseudotime as the priority time with the neighbourhood graph as the input of `omicverse.utils.cal_paga`.

(5) The number of noise clusters, we used `scanpy.tl.leiden` in scanpy to perform unsupervised clustering on the generated single-cell profiles, with the resolution set to 1.0, and we identified the categories with less than 25 cells after clustering as noisy clusters and counted the number of noisy clusters as an assessment of the generation quality.

(6) Density assessment of pseudotime, after we obtained the pseudotime of single-cell profiles using pyVIA as described previously, we assessed the variance of the pseudotime of target interpolated cells as one of the metrics for the assessment of developmental trajectory reconstruction

- To further demonstrate the efficacy of BulkTrajBlend, the authors could consider applying it to existing datasets that have both bulk and single cell data. By simulating varying conditions of cell dropout, the authors could showcase BulkTrajBlend's ability to accurately discern cell*

transitions between different states.

Response: We agree that the simulating varying conditions of cell dropout. We appreciate the reviewer's suggestion, and we agree with the importance of demonstrating the efficacy of BulkTrajBlend in practical scenarios. To address this, in the simulated "interruption" dataset, we strategically deleted a part of specific cell types, such as Ngn^{High} EP cells in the mouse pancreatic development and differentiation dataset, HSC 2 cells in the bone marrow differentiation dataset, and Immature neurons in the dentate gyrus neurons. This deliberate manipulation ensured the introduction of disruptions in the original developmental trajectory of the cells. Subsequently, we conducted tests to evaluate BulkTrajBlend's capability to reconstruct the developmental trajectory after interpolating these "interrupted" cells. The results demonstrate that BulkTrajBlend successfully restores the developmental trajectory, state transfer probability, and retains a low variant phenotype, showcasing its effectiveness under varying conditions of cell dropout in result "Proficient Reconstruction of Cell Developmental Trajectories in Simulated "interruptions" Single-Cell Profiles".

Within the mouse pancreatic development dataset, PAGA plots illustrated a baseline probability of 0.04 for Ngn3^{High} EP cells differentiating into Pre-endocrine cells. In the corresponding "interruptions" dataset, this probability was 0. BulkTrajBlend interpolation increased the probability to 0.035 (Fig.3d-3g, Extend Data Fig.6a-6c). In the mouse dentate gyrus neurons development, Granule Immature cells had baseline differentiation probability to Granule Mature cells of 0.018, while no probability was observed in simulated "interruptions" dataset. BulkTrajBlend's interpolation resulted in a probability elevation to 0.019 (Fig.3g, Extend Data Fig.5d-5f, Extend Data Fig.6d-6f). In human bone marrow development, hematopoietic stem cells stage 2 (HSC 2) cells showed a differentiation probability into monocytes of 0.082, compared to 0 in the simulated "interruptions" dataset. Following BulkTrajBlend interpolation, the probability rose to 0.079 (Fig.3g, Extend Data Fig.5j-5l, Extend Data Fig.6g-6i). Notably, the original pseudotime variability in the three datasets was preserved after interpolation. These analyses collectively highlight BulkTrajBlend's effectiveness in accurately reconstructing authentic developmental trajectories.

Benchmarking and Parameter Sensitivity:

- 5. The authors should rigorously benchmark BulkTrajBlend against other approaches employed in single cell studies, specifically those utilizing either the Variational Autoencoder (VAE) or Generative Adversarial Network (GAN) frameworks as well as other bulk-deconvolution approaches.*

Response: We appreciate the reviewer's emphasis on the need for rigorous benchmarking against other approaches in the field of bulk-deconvolution. It is crucial to note that the majority of existing bulk-deconvolution algorithms primarily focus on estimating cell proportions or feature vectors. Algorithms directly deconvolving bulk RNA-seq to generate scRNA-seq are scarce, with bulk2space being one of the reported algorithms, also based on the core principle of beta-VAE. To address this, we conducted a thorough comparison within the BulkTrajBlend framework, examining the impact of different generative principles, including CGAN and ACGAN, both capable of specifying categories for generation. Our findings indicate that

utilizing the Beta-VAE model within BulkTrajBlend consistently yields superior results in terms of marker correlation, similarity, state transitions, and cell variant profiles after interpolation.

To gauge the efficacy and accuracy of BulkTrajBlend in the context of cell differentiation trajectory recovery, a rigorous benchmarking exercise is undertaken. The VAE module within BulkTrajBlend is systematically compared against alternative generative models, including conditional generative adversarial networks (CGAN) and auxiliary conditional GANs (ACGAN). The benchmarking process encompasses the evaluation of various performance metrics, encompassing the correlation of cell-type marker gene expression, marker gene similarity (quantified via cosine similarity), probability of trajectory conversion subsequent to interpolation, and the degree of data variability following interpolation.

6. *Given the potential vastness of the parameter space associated with this method, a comprehensive benchmarking study is essential. The authors should test a diverse set of parameters to determine how variations might significantly influence the results.*

Response: We express our gratitude to the reviewer for highlighting the importance of conducting a comprehensive benchmarking study on parameter variations. In response to this valuable suggestion, the revised manuscript extensively explores hyperparameters from three critical perspectives. Firstly, we scrutinized the influence of the input single-cell data size, varying it from 1,000 to 20,000. Subsequently, we rigorously assessed the dimension of the interpolation size, spanning from 1x to 10x of the original cell count. Last, we systematically evaluated the number of neurons in BulkTrajBlend's hidden layer, ranging from 64 to 1024. The detailed findings on the optimal parameter space have been incorporated into the results section, providing comprehensive insights into the impact of variations in hyperparameters. We present what we consider to be the optimal parameter space at the end of result section:

In conclusion, the ideal hyperparameter setting involve using the entire single-cell dataset, interpolating at a scale of 2x or 4x, and a hidden layer configured with 64 neurons. Under these optimal hyperparameters, BulkTrajBlend effectively reconstructed the nIPC-OPC developmental flow pattern in dentate gyrus datasets and the HSC-Basophil flow pattern in hematopoietic system development datastes (Fig.2m-2n).

Addressing Potential False Positives:

7. *Another concern for BulkTrajBlend to produce false positives. For instance, in mature tissues, certain cell states might lack transitional phases. How does BulkTrajBlend handle such scenarios? Does it erroneously predict "interrupted" cells? A quantitative exploration of this aspect is imperative for the tool's broader acceptance and trustworthiness.*

Response: We sincerely appreciate the reviewer's insightful comment regarding the potential for false positives in BulkTrajBlend. We introduced the AE model to estimate cell proportion in the new BulkTrajBlend, which firstly constructs Bulk RNA-seq with cell proportion labels from scRNA-seq, and the cell proportion is used as the output of the encoder therein.

Immediately after that, we then use Bulk RNA-seq as the input of the optimal AE model, and since the input of the encoder is the same as the output of the generator, then after real Bulk RNA-seq is used as the input of the encoder, the output of the encoder is the cell proportions of real Bulk RNA-seq. We selected a Bulk RNA-seq dataset from the hippocampus to verify this conclusion¹³. The dataset used next-generation RNA sequencing (RNA-seq) to produce a quantitative, whole genome atlas of gene expression for every excitatory neuronal class in the hippocampus; namely, granule cells and mossy cells of the dentate gyrus, and pyramidal cells of areas CA3, CA2, and CA1. **Based on BulkTrajBlend's cell ratio prediction, we found that granule cells were predominantly immature granule and mature granule in Bulk RNA-seq, and mossy cells were predominantly mossy in Bulk RNA-seq. whereas in CA1, CA2, CA3, the ratio of granule to mossy was close (Response Fig.1), which is consistent with the actual distribution of neurons in the mouse hippocampal region.** Therefore, in the new BulkTrajBlend will effectively identify the proportion of different types of cells.

The potential false-positive rate is a concern, and we introduced the error overlap rate as a judgement of false positives of predicted cells, which is based on the following assumptions:

- 1, we assumed that a particular type of cell deconvolved from Bulk has a steady state, and its particular cell in a non-steady state is regarded as a transition cell
- 2, We assume that the transition cells have the characteristics of both types of cells and are seen as overlapping communities in terms of community, and that this fraction of transition cells can be captured using the GNN.

We define False Overlap Rate (FOR) as the judgement of false positive rate, defined as follows: if the overlapping community OC, in which a certain type of cell is located, is not contained in his unique original community UC, we denote it as False Overlap FO, and the formula of False Overlap Rate FOR is defined as follows:

$$FOR = \frac{Number_{FO}}{Number_{OC}}$$

We described the FOR in Supplementary Note 2:

We found that the FOR of nIPC, Cck-TOX, and Neuroblast is higher than 0.3, while the OPC we used for interpolation is 0.21, indicating that wrongly overlapping cells only account for 20% of the overlapping communities, which we used for back-interpolation (Extended Data Fig.1g).

Response to Review #2 Fig 1 Neuronal fraction predicted in the hippocampus. The horizontal coordinate represents the sample name for Bulk RNA-seq and the vertical coordinate represents the percentage of cells predicted.

Reference

- 1 Wu, H., Kirita, Y., Donnelly, E. L. & Humphreys, B. D. Advantages of single-nucleus over single-cell RNA sequencing of adult kidney: rare cell types and novel cell states revealed in fibrosis. *Journal of the American Society of Nephrology: JASN* **30**, 23 (2019).
- 2 Mereu, E. *et al.* Benchmarking single-cell RNA-sequencing protocols for cell atlas projects. *Nature Biotechnology* **38**, 747-755, doi:10.1038/s41587-020-0469-4 (2020).
- 3 Denyer, T. & Timmermans, M. C. P. Crafting a blueprint for single-cell RNA sequencing. *Trends Plant Sci* **27**, 92-103, doi:10.1016/j.tplants.2021.08.016 (2022).
- 4 Thibivilliers, S., Anderson, D. & Libault, M. Isolation of Plant Root Nuclei for Single Cell RNA Sequencing. *Curr Protoc Plant Biol* **5**, e20120, doi:10.1002/cppb.20120 (2020).
- 5 Gao, C., Zhang, M. & Chen, L. The comparison of two single-cell sequencing platforms: BD rhapsody and 10x genomics chromium. *Current genomics* **21**, 602-609 (2020).
- 6 Frishberg, A. *et al.* Cell composition analysis of bulk genomics using single-cell data. *Nature methods* **16**, 327-332, doi:10.1038/s41592-019-0355-5.
- 7 Wang, X., Park, J., Susztak, K., Zhang, N. R. & Li, M. Bulk tissue cell type deconvolution with multi-subject single-cell expression reference. *Nature communications* **10**, 380 (2019).
- 8 Higgins, I. *et al.* beta-VAE: Learning Basic Visual Concepts with a Constrained Variational Framework. (2016).
- 9 Lotfollahi, M., Wolf, F. A. & Theis, F. J. scGen predicts single-cell perturbation responses. *Nature Methods* **16**, 715-721, doi:10.1038/s41592-019-0494-8 (2019).
- 10 Traag, V. A., Waltman, L. & Van Eck, N. J. From Louvain to Leiden: guaranteeing well-connected communities. *Scientific reports* **9**, 1-12 (2019).
- 11 Yang, J. & Leskovec, J. Structure and overlaps of ground-truth communities in networks. *ACM Transactions on Intelligent Systems and Technology (TIST)* **5**, 1-35 (2014).
- 12 Shchur, O. & Günnemann, S. Overlapping community detection with graph neural networks. *arXiv preprint arXiv:1909.12201* (2019).
- 13 Cembrowski, M. S., Wang, L., Sugino, K., Shields, B. C. & Spruston, N. Hipposeq: a comprehensive RNA-seq database of gene expression in hippo campal principal neurons. *eLife* **5**, e14997, doi:10.7554/eLife.14997 (2016).

REVIEWER COMMENTS

Reviewer #1 (Remarks to the Author):

The updated manuscript "OmicVerse: A single pipeline for exploring the entire transcriptome universe" by Zeng et al. addresses many of the criticisms from the first round of reviews. However, some of the new text is unclear and some of the previous points are insufficiently addressed.

A major concern was the lack of definition of the term "interrupted cells". The authors addressed this concern by adding the following sentence:

> A recognized impediment in single-cell sequencing is the 'interruption' -- the omission of certain cell types due to technological constraints on the sequencing platform and interruption the trajectory of cell differentiation, such as the enzymatic lysis-related loss of podocytes and intercalated cells¹, the differentiation from HPC to podocytes was interrupted, and the filtering-induced absence of neutrophils, cardiomyocytes, neuronal cells, and megakaryocytes and the differentiation from nIPC to neurons was interrupted²⁻⁴

While I appreciate the authors' attempt to clarify the concept, the assertion that "A recognized impediment in single-cell sequencing is the 'interruption'" is problematic. To my knowledge, the term 'interruption' has not been used in this context before.

The authors are right to say that single-cell sequencing can miss or "omit" interesting cells, and they provide sufficient references to previous literature to make that point. Yet, the phrase "interruptions" does not convey this concept.

I recommend that the authors remove the phrase "interrupted" and replace it with something more evocative, like `_omitted cells_` or `_missing cells in a trajectory_`.

A new major concern about the updated manuscript is the presentation of the BulkTrajBlend algorithm.

The authors significantly expanded the description of the algorithm. However, from the description in the section "Design concept of BulkTrajBlend and Benchmarking" (line 74-134), the input and output of the algorithm and the novelty of the method remain unclear.

For example, the authors describe one of the enhancements of the method as "inspired by TAPE" (line 84). However, there is no additional explanation of what TAPE does, nor is it mentioned again in the method section.

The authors describe that they "[utilize] ground truth bulk RNA-seq as input for calculating the true fraction of cells", but it is unclear how.

To improve this section, the authors should rewrite the section into two bits: first, a traditional Introduction section that explains the problem in more detail, how existing methods like TAPE infer the cell type proportions from bulk RNA-seq data, why the beta-variational autoencoder is a useful model in this context. The Results section can then contain an explanation what novel approaches or combination of methods BulkTrajBlend implements to resolve some of the problems identified in the Introduction.

The problem with the current presentation is that it is unclear why the authors in addition to the beta-VAE need a clustering step, another VAE. These steps are merely motivated with a reference to "this approximation introduces a level of noise and bias" (line 95) and the "unconstrained nature of the decoder's output" (line 98). Yet, without additional data it is difficult to verify that these steps are necessary.

In line 116, the authors state that "real communities are overlapping". This claim is overly broad and should be adjusted. Non-overlapping definitions of cell types have worked well as concepts for analyzing single-cell data. This does not mean that using a definition where communities overlap is not useful, but it is only one among many valid approaches to model the data.

In line 125, the authors refer to conditional GANs and auxiliary conditional GANs, but again fail to

provide a reference for these methods.

Lastly, given the increased emphasis of the manuscript on the BulkTrajBlend algorithm, the authors should consider changing the title of the manuscript to a title that reflects this focus.

On point 2

What is the "state space"?

On point 3

Sounds good.

On point 8 (minor concerns)

I thank the authors for clarifying why they refer to UMAP to find the neighborhood graph. The terminology of scanpy is confusing as the comment on Github at <https://github.com/scverse/scanpy/issues/277#issuecomment-427333649> explains. Under the hood, the method calls pynndescent (<https://github.com/lmcinnes/pynndescent>). The authors should update the text to reflect this.

On point 9 (minor concerns)

The additional context is helpful. However, the notation is not precise. The set of real numbers is commonly written as \mathbb{R} .

In general, the text describing the beta-variational autoencoder is uncomfortably close to the original description in Higgins (2017). Here is a paragraph from Higgins et al.:

> Let $D = \{X, V, W\}$ be the set that consists of images $x \in \mathbb{R}^N$ and two sets of ground truth data generative factors: conditionally independent factors $v \in \mathbb{R}^K$, where $\log p(v|x) = \sum_k \log p(v_k|x)$; and conditionally dependent factors $w \in \mathbb{R}^H$. We assume that the images x are generated by the true world simulator using the corresponding ground truth data generative factors: $p(x|v, w) = \text{Sim}(v, w)$.

And this is the corresponding paragraph from Zeng et al.:

> Given a set $D = X, V, W$, where $x \in X$ represents gene expression vectors, $v \in V$ represents cell type proportions, satisfying $\log p(v|x) = \sum \log p(v_k|x)$, where $v \in \mathbb{R}^K$; and $w \in W$ represents conditionally correlated generative factors. We assume that gene expression vectors x are generated by a real-world simulator S , with the corresponding generative factors as input, i.e., $p_\theta(x|v, w) = S(v, w)$, where θ represents the generative model parameters.

Such similarities are present for the full section line 584-636. In my opinion, the problems with this section are two-fold:

1. The lack of context can make it easy for the reader to miss that this is a lightly modified copy of Higgins' method section. I recommend condensing the section to the essentials and explicitly referring the readers to the original work.
2. Some of the notational changes are mathematically wrong. For example, Higgins et al. introduce $x \in \mathbb{R}^N$. This was changed to $x \in X$, which is mathematically not meaningful as X is not a set. This poses a barrier for readers who are not already familiar with beta-variational autoencoders to understand what is done on a technical level.

Reviewer #3 (Remarks to the Author):

My major concern is that the BulkTrajBlend algorithm is poorly described. If I understand

correctly, the authors are trying to generate "interrupted" single cell gene expression profiles. While this is an important question to be addressed in the single cell field, I am very confused about the detailed workflow described in Figure 1.

1. How exactly does beta-VAE work? It seems that beta-VAE generates X (single cell gene expression) from V (cell type fraction) and W (correlated generative factor). But the question is how was the beta-VAE trained? Maybe the authors took a bulk expression profile as the input, forced μ and θ to represent V and W of a specific single cell, and reconstructed the expression of that specific single cell? Also what is the "correlated generative factor"? The authors should define it.

2. How does AE and beta-VAE relate to each other? I understand that AE tries to encode cell type fractions in the bottleneck layer. Maybe the authors took cell type fractions predicted from a real bulk sample, as the beta-VAE V, for generating single cell expression profiles?

3. How does the filtering work? Maybe the authors took single cell profiles generated from beta-VAE for clustering analysis, and then removed poorly-classified cells?

4. What does "co-CellType" mean in Figure 1B?

Several minor comments:

1. Typos, such as "similuar" in line 558, should be corrected.

2. Not sure how does reference 22 relate to the formulation of "interrupted" single cells.

Reviewer #4 (Remarks to the Author):

Point-to-point replies to the reviewers' comments:

In the following, we present our response to the reviewers' comments. We give *comments (blue)*, *point-by-point answers (green)* to the questions, and in parts copy parts of the text or specific panels (black), which directly correspond to comments or reference to them. All changes in the manuscript text file with *colour highlighting*.

We express our gratitude to the esteemed reviewers for their valuable comments, recommendations, and constructive feedback.:

- 1) **Redescribes how BulkTrajBlend works.** In the revised manuscript, we have restated the algorithmic structure and workings of BulkTrajBlend, and also added the roles that algorithms such as AE, beta-VAE, GNN, and so on, play in the model, as well as revising the exposition of the formulas in the manuscript in a more precise and concise manner.

Reviewer#1:

The updated manuscript "OmicVerse: A single pipeline for exploring the entire transcriptome universe" by Zeng et al. addresses many of the criticisms from the first round of reviews. However, some of the new text is unclear and some of the previous points are insufficiently addressed.

Major Concerns

1. *A major concern was the lack of definition of the term "interrupted cells". The authors addressed this concern by adding the following sentence:*

> A recognized impediment in single-cell sequencing is the 'interruption' -- the omission of certain cell types due to technological constraints on the sequencing platform and interruption the trajectory of cell differentiation, such as the enzymatic lysis-related loss of podocytes and intercalated cells1, the differentiation from HPC to podocytes was interrupted, and the filtering-induced absence of neutrophils, cardiomyocytes, neuronal cells, and megakaryocytes and the differentiation from nIPC to neurons was interrupted2-4

While I appreciate the authors' attempt to clarify the concept, the assertion that "A recognized impediment in single-cell sequencing is the 'interruption'" is problematic. To my knowledge, the term 'interruption' has not been used in this context before.

The authors are right to say that single-cell sequencing can miss or "omit" interesting cells, and they provide sufficient references to previous literature to make that point. Yet, the phrase "interruptions" does not convey this concept.

I recommend that the authors remove the phrase "interrupted" and replace it with something more evocative, like _omitted cells_ or _missing cells in a trajectory_.

Response: We are grateful for the reviewer's perceptive observations and guidance. To enhance the clarity of the terminology used, we agree that it is more accurate to describe the cell types

that are not captured in sequencing as “omitted cells.” We will carefully amend the term “interrupted cells” to “omitted cells” throughout our manuscript. This change will more precisely communicate the concept of potentially missed cell populations and address the issue raised. We value the reviewer’s input, which helps us refine our narrative and ensure that our terminology aligns with the established scientific discourse. Thank you for assisting us in improving the precision and quality of our work.

2. *A new major concern about the updated manuscript is the presentation of the BulkTrajBlend algorithm.*

The authors significantly expanded the description of the algorithm. However, from the description in the section "Design concept of BulkTrajBlend and Benchmarking" (line 74-134), the input and output of the algorithm and the novelty of the method remain unclear.

For example, the authors describe one of the enhancements of the method as "inspired by TAPE" (line 84). However, there is no additional explanation of what TAPE does, nor is it mentioned again in the method section. The authors describe that they "[utilize] ground truth bulk RNA-seq as input for calculating the true fraction of cells", but it is unclear how.

To improve this section, the authors should rewrite the section into two bits: first, a traditional Introduction section that explains the problem in more detail, how existing methods like TAPE infer the cell type proportions from bulk RNA-seq data, why the beta-variational autoencoder is a useful model in this context. The Results section can then contain an explanation what novel approaches or combination of methods BulkTrajBlend implements to resolve some of the problems identified in the Introduction.

Response: We are grateful to the reviewer for the constructive feedback and agree that the description of the BulkTrajBlend algorithm needs further clarification. We have revised our manuscript to present a clearer overview of the problem, existing methods, and our novel contributions to the field.

In the “Introduction” section, we now elucidate the challenges involved in deconvolving bulk RNA-seq data into cell-type proportions, with a brief review of techniques currently employed, such as TAPE, as well as the limitations they present. We discuss why advanced models like beta-variational autoencoders (VAE) hold promise in addressing these challenges, providing readers with a solid foundation on the topic.

Conversely, bulk RNA-seq of whole tissues intrinsically includes these 'omitted' cells. It should be acknowledged that there is no existing algorithm that can directly solve the "omitted" cell problem, but similar to this problem, there are some deconvolution algorithms, such as TAPE, CIBERSORT (CS), MuSiC, CIBERSORTx (CSx), Bisque etc. These algorithms do not really solve the "omitted" cell problem because they lack a generative capability, which suggests that Generative Adversarial Networks (GANs) may be the best solution to the "omitted" cell problem. "omitted" cell problem, which suggests that Generative algorithms like Generative Adversarial Networks (GANs) or Variational Autoencoder (VAE) may be a potential way to solve the "omitted" cell problem.

Moving to the “Results” section, we have added a more detailed account of how the TAPE algorithm has informed our methodological approach. We clarify the process by which we simulate bulk RNA-seq from real single-cell RNA-seq data, using the obtained cell fraction vector from single-cell RNA-seq as the basis for our Encoder’s output and the Decoder’s input. This setup allows us, when applying the Encoder to actual bulk RNA-seq data, to derive the cell fraction vectors corresponding to the makeup of the bulk sample. This detailed description should make it evident how our approach refines and extends upon existing methodologies to better approximate the true fractions of cells in a bulk RNA-seq sample.

BulkTrajBlend advances the foundational structure of autoencoders (AE) and β -VAE. These enhancements encompass (1) employing an AE to construct a Bulk RNA-seq generator analogous to real Bulk RNA-seq inspired by TAPE, we modelled the cellular proportion space of Bulk RNA-seq on the output of the Encoder, the input of the Decoder. Subsequently utilizing ground truth bulk RNA-seq generated from single cell RNA-seq as input of Encoder for calculating the true cellular proportion.

It should be added that we elaborate on the implementation of this part of TAPE in the Method section on page 21.

(1) Cell fraction calculation:

To estimate the proportion of cells in Bulk RNA-seq, we first annotated the single-cell data with cell types and summed the gene counts of single cells of different cell types by cell to obtain an $N \times M$ matrix, where M represents the number of cell types and N represents the number of genes. We define this $N \times M$ matrix as the simulated Bulk RNA-seq cell type matrix, and then we sum M columns of each row to get the simulated Bulk RNA-seq $B_{\text{simulated}}$, and we input the simulated Bulk RNA-seq into the self-encoder. In the self-encoder, we define the output of the encoder as T , and we make T close to $(\text{Number of the cell})/(\text{Number of all cells})$, i.e., Cell Fraction, by training AE. We then define the output of the generator as G and we make G and $B_{\text{simulated}}$ close to each other by MAE as an evaluation. After training the optimal AE, we change the input to real Bulk RNA-seq $B_{\text{groundtruth}}$, at which time the output of the encoder, T , is the Cell Fraction corresponding to real Bulk, which we use as the range of the generator space for the subsequent beta-VAE.

- The problem with the current presentation is that it is unclear why the authors in addition to the beta-VAE need a clustering step, another VAE. These steps are merely motivated with a reference to "this approximation introduces a level of noise and bias" (line 95) and the "unconstrained nature of the decoder's output" (line 98). Yet, without additional data it is difficult to verify that these steps are necessary.*

Response: We are thankful for the reviewer’s discerning point. During the process of single-cell data generation and clustering with VAE, we encountered the emergence of numerous small cell clusters, each typically comprising 15-20 cells. These clusters, post-dimensionality reduction, were found to be evenly scattered throughout the space. To address this, we employed unsupervised clustering coupled with a size threshold to identify and remove these noisy cells. These noisy cells may have potential biological significance, but they do not help us recover "omitted" cells, so we choose to filter them rather than retain them in subsequent analyses. We

show this noisy phenomenon in the extended data Fig 1e, with additional descriptions in the corresponding places in the line 112.

It is noteworthy that this approximation introduces a level of noise and bias into the generated data (Extended Data Fig. 1e). Consequently, unsupervised clustering is employed as a data refinement strategy to mitigate the impact of noise and enhance data robustness. We use unsupervised clustering to filter out "noisy" cells, which are identified using community size.

4. *In line 116, the authors state that "real communities are overlapping". This claim is overly broad and should be adjusted. Non-overlapping definitions of cell types have worked well as concepts for analyzing single-cell data. This does not mean that using a definition where communities overlap is not useful, but it is only one among many valid approaches to model the data.*

Response: We are thankful to the reviewer for their critical feedback and for drawing our attention to the breadth of the claim made in line 116. In light of your comments, we agree that stating "real communities are overlapping" without appropriate nuance does not fully represent the wide spectrum of approaches used for analyzing single-cell data, where non-overlapping models can indeed be effective and have frequently been used.

Therefore, we have revised the manuscript to temper this assertion. We now acknowledge that the use of non-overlapping cell type definitions has been successfully applied in the field and that our proposed method incorporating the concept of overlapping communities is but one of several valid strategies for modeling single-cell data. Our motivation for exploring this direction stemmed from our aim to recover not only the "omitted" cells but also to identify cells that may bridge the identified populations, thereby potentially enhancing the continuity of our reconstructed trajectories.

To better articulate the implications of this approach, we have included further comparisons within a new figure (Responding to Review #1 Fig 1). This illustration contrasts the effects of reconstructing trajectories by restoring solely the "omitted" cells versus incorporating both "omitted" and contiguous overlapping cells. We have also refined the corresponding text within the manuscript to reflect this more considered perspective.

Given that BulkTrajBlend's primary objective is to interpolate data from original scRNA-seq data, the focus shifts to the targeted extraction of cells from the generated single-cell data. Considering the inherent challenges associated with cell annotation, the input single-cell data encompassing diverse cell types is expected to exhibit overlaps in real-world scenarios. The "omitted" cells we need to recover should be with the continuous state of the cells, the traditional community discovery algorithms can not identify the overlapping cell communities, and the "omitted" cells generated by beta-VAE are directly transferred to the "omitted" cells, which will lose the continuous state of the cells. cell generated by β -VAE is directly restored to the original

single-cell data, which will lose the continuous state of the cell. To solve this problem, we introduce NOCD, a GNN-based algorithm for identifying overlapping communities and achieves the best level in the existing baseline³⁰. Using NOCD enabling the identification of overlapping cell communities. We also used the "omitted" cell and its "omitted" cell in the overlapping community state as the target cells for recovery. This insight is integral to the subsequent task of recovering and reconstructing cell differentiation trajectories within the single-cell sequencing data (Fig.1b).

Response to Review #1 Fig 1 | Restoration of "omitted" cell continuity by overlapping communities. (a) UMAP shows, from left to right, the cells before the insertion of the "omitted" cells, the "omitted" cells only, the "omitted" cells and the cells in their overlapping neighbourhoods for Dentate Gyrus Neurogenesis. (b) Force-Directed Graph shows from left to right, the cells before the insertion of the "omitted" cells, the "omitted" cells only, the "omitted" cells and the cells in their overlapping neighbourhoods for Dentate Gyrus Neurogenesis, Color-Coded by Cell Type.

5. *In line 125, the authors refer to conditional GANs and auxiliary conditional GANs, but again fail to provide a reference for these methods.*

Response: We are grateful to the reviewer for highlighting this omission. We have now included citations for both conditional GANs and auxiliary conditional GANs in line 148-149 of the revised manuscript.

6. *Lastly, given the increased emphasis of the manuscript on the BulkTrajBlend algorithm, the authors should consider changing the title of the manuscript to a title that reflects this focus.*

Response: Thank you for your helpful feedback concerning the title of our manuscript. We

value your consideration of the focus given to the BulkTrajBlend algorithm. Following your insightful recommendation, we have updated the title to more precisely reflect the distinctive contributions highlighted in our work. We suggest the revised title to be:

“OmicVerse: A Framework for Bridging and Deepening Insights Across Bulk and Single-Cell Sequencing”

We contend that this refined title captures the essential functions of the framework we developed, particularly emphasizing the pivotal role of the BulkTrajBlend algorithm. Bridging serves to denote the algorithm’s capability to integrate “omitted” cells from bulk RNA-seq into single-cell RNA-seq analyses, a feature that is the focus of our paper’s first half. In addition, we opted for Deepening Insights to convey our commitment to facilitating extensive exploration of the entire transcriptome, which is presented in the second half of the paper and within the supplementary materials. This holistic analysis was rendered feasible by refactoring various existing R packages for transcriptome analysis into a singular, streamlined framework. Furthermore, we standardized the data formats across different algorithms, enabling users to delve deep into data analysis without interruption from the varying runtime limitations associated with disparate algorithms.

7. *### On point 2: What is the "state space"?*

Response: I'm sorry that our definition of state space has confused you, in fact, we define the high-dimensional features learned by different LAYERS in the Encoder as the state space of Bulk RNA-seq.

8. *### On point 3: Sounds good.*

Response: Thanks for the compliment.

9. *### On point 8 (minor concerns). I thank the authors for clarifying why they refer to UMAP to find the neighborhood graph. The terminology of scanpy is confusing as the comment on Github at <https://github.com/scverse/scanpy/issues/277#issuecomment-427333649> explains. Under the hood, the method calls pynndescent (<https://github.com/lmcinnes/pynndescent>). The authors should update the text to reflect this.*

Response: We gratefully acknowledge the reviewer’s guidance on the intricacies of the terminology concerning scanpy’s neighborhood graph computation. We recognize, based on the Github discussion, that scanpy utilizes the pynndescent library for this purpose. We have updated the pertinent sections of our manuscript to correctly represent this method. Our thanks extend to the reviewer for allowing us to refine our manuscript with this important correction.

(3) Computation of single-cell neighborhood graph

Here, we used the scanpy.pp.neighbors function from Scanpy to compute the cell neighborhood

graph. For detailed mathematical descriptions, please refer to the relevant papers and documentation of nearest neighbor descent in Scanpy and PyNNDescent58.

10. ### On point 9 (minor concerns). The additional context is helpful. However, the notation is not precise. The set of real numbers is commonly written as \mathbb{R} .

In general, the text describing the beta-variational autoencoder is uncomfortably close to the original description in Higgins (2017). Here is a paragraph from Higgins et al.:

> Let $D = \{X, V, W\}$ be the set that consists of images $x \in \mathbb{R}^N$ and two sets of ground truth data generative factors: conditionally independent factors $v \in \mathbb{R}^K$, where $\log p(v|x) = \sum_k \log p(v_k|x)$; and conditionally dependent factors $w \in \mathbb{R}^H$. We assume that the images x are generated by the true world simulator using the corresponding ground truth data generative factors: $p(x|v, w) = \text{Sim}(v, w)$.

And this is the corresponding paragraph from Zeng et al.:

> Given a set $D = \{X, V, W\}$, where $x \in X$ represents gene expression vectors, $v \in V$ represents cell type proportions, satisfying $\log p(v|x) = \sum \log p(v_k|x)$, where $v \in \mathbb{R}^K$; and $w \in W$ represents conditionally correlated generative factors. We assume that gene expression vectors x are generated by a real-world simulator S , with the corresponding generative factors as input, i.e., $p(x|v, w) = S(v, w)$, where θ represents the generative model parameters.

Such similarities are present for the full section line 584-636. In my opinion, the problems with this section are two-fold:

1. The lack of context can make it easy for the reader to miss that this is a lightly modified copy of Higgins' method section. I recommend condensing the section to the essentials and explicitly referring the readers to the original work.
2. Some of the notational changes are mathematically wrong. For example, Higgins et al. introduce $x \in \mathbb{R}^N$. This was changed to $x \in X$, which is mathematically not meaningful as X is not a set. This poses a barrier for readers who are not already familiar with beta-variational autoencoders to understand what is done on a technical level.

Response: We appreciate the reviewer's insightful comment. We are sorry that the uncomfortably similar part confused you. We have subtracted the content of the similar part and cited the original papers. At the same time, the formula containing the real set is replaced by \mathbb{R} and checked for mathematical meaning.

(2) Generation of single-cell data:

Given a dataset $\{X, V, W\}$, where the vector $x \in \mathbb{R}^M$ in the gene expression matrix X represents gene expression vector of a cell, the vector $v \in \mathbb{R}^K$ in the matrix V represents cell type proportion, satisfying $\log p(v|x) = \sum_k \log p(v_k|x)$, where $v \in \mathbb{R}^K$ is restricted by a loss function:

$$\text{MAE} = \sum_v |v - \hat{v}|$$

Here \hat{v} is the predicted proportions of certain cell type.

The vector $w \in \mathbb{R}^K$ in the matrix W represents conditionally correlated generative factor. The factor w is obtained from the same class of cells through the beta-VAE Encoder. For each class of cells, the average value after model training represents a class of cell-specific w , and it is not restricted by adding a loss function. According to Higgins et al²⁶, we hypothesize that gene expression vectors x are generated by a probability model $p_\theta(x|v, w)$, where θ represents the generative model parameters. The model learns the joint distribution of the data x and a set of latent variables z ($z \in \mathbb{R}^M$, where $M \geq K$) for generating observed data x , i.e., $p_\theta(x|z) \approx p(x|v, w)$, and approximates the true posterior distribution $p_\theta(z|x)$ with an approximate posterior distribution $q_\phi(z|x)$ that is easier to compute. Our goal is to ensure that the inferred latent variables z capture the generative factors w in a disentangled manner. A disentangled representation implies that individual latent unit is sensitive to variations in a single generative factor while being relatively invariant to variations in other factors. In a disentangled representation, knowledge of one factor can be generalized to new configurations of other factors. The conditionally correlated generative factors w can remain entangled in a separate subset of z and are not used to represent v .

Reviewer #3 (Remarks to the Author):

My major concern is that the BulkTrajBlend algorithm is poorly described. If I understand correctly, the authors are trying to generate "interrupted" single cell gene expression profiles. While this is an important question to be addressed in the single cell field, I am very confused about the detailed workflow described in Figure 1.

1. *How exactly does beta-VAE work? It seems that beta-VAE generates X (single cell gene expression) from V (cell type fraction) and W (correlated generative factor). But the question is how was the beta-VAE trained? Maybe the authors took a bulk expression profile as the input, forced μ and θ to represent V and W of a specific single cell, and reconstructed the expression of that specific single cell? Also what is the "correlated generative factor"? The authors should define it.*

Response: We are grateful to the reviewer for their comprehensive questions about the beta-VAE in our BulkTrajBlend algorithm. Recognizing that the original manuscript lacked a detailed explanation of the beta-VAE training, we regret any resultant misunderstanding.

The beta-VAE is trained to generate single-cell gene expression data (X) from cell type fractions (V) and a correlated generative factor (W). We have methodically defined V and incorporated it into the loss function to guide the generation of data that aligns with expected cell type proportions—a constraint designed to ensure the biological plausibility of the data.

W serves as a latent feature representing traits unique to each cell class. We obtain W by

averaging the output of the beta-VAE's Encoder across each class of homogeneous cells, after training concludes.

Notably, we have opted not to apply a specific loss function for W , preferring to allow these features to form as a byproduct of the training. The revised manuscript expands on this topic, detailing the roles of V and W in our model to provide the reader with a comprehensive understanding of our approach. Below is the revised main text

BulkTrajBlend advances the foundational structure of autoencoders (AE) and β -VAE. These enhancements encompass (1) employing an AE to construct a Bulk RNA-seq generator analogous to real Bulk RNA-seq inspired by TAPE, we modelled the cellular proportion space of Bulk RNA-seq on the output of the Encoder, the input of the Decoder. Subsequently utilizing ground truth bulk RNA-seq generated from single cell RNA-seq as input of Encoder for calculating the true cellular fractions. (2) When we trained beta-VAE using real single cell RNA-seq, the Encoder outputs were V (cell type fraction) and W (cell type correlated generative factor), and we added a loss function for minimising the relationship between V and the real cell type fraction. We obtained W for each cell at the end of model training and averaged W for each cell type to represent that cell type. (3) We used the true cell type fraction V calculated by AE with the cell type-associated generating factor W obtained by beta-VAE as input to beta-VAE for generating single-cell data and deploying unsupervised clustering to denoise and refine the outcomes of the β -VAE.

2. *How does AE and beta-VAE relate to each other? I understand that AE tries to encode cell type fractions in the bottleneck layer. Maybe the authors took cell type fractions predicted from a real bulk sample, as the beta-VAE V , for generating single cell expression profiles?*

Response: We thank the reviewer for their valuable comment. In our BulkTrajBlend algorithm, autoencoders (AE) and beta-variational autoencoders (beta-VAE) are separate components that function synergistically.

The beta-VAE utilizes cell type proportions, as quantified by the AE, for its generative input. We recognize that our original manuscript did not adequately detail this interaction; hence, we have amended the text to rectify this oversight.

In the “Results” section, particularly within the “Design Concept of BulkTrajBlend and Benchmarking,” we describe how the AE is initially applied to model cell proportions from bulk RNA-seq data. In this context, the AE treats the cell ratio as the output of the Encoder and as input for the Decoder. Further training of the AE involves iterative refinement using bulk RNA-seq data simulated from single-cell sources. These steps are now more thoroughly explained in the revised manuscript, ensuring clear communication of how AE and beta-VAE contribute to our analytical pipeline, which is expressed as follows in the main text:

Result: BulkTrajBlend advances the foundational structure of autoencoders (AE) and β -VAE. These enhancements encompass (1) employing an AE to construct a Bulk RNA-seq generator

analogous to real Bulk RNA-seq inspired by TAPE, we modelled the cellular proportion space of Bulk RNA-seq on the output of the Encoder, the input of the Decoder. Subsequently utilizing ground truth bulk RNA-seq generated from single cell RNA-seq as input of Encoder for calculating the true cellular proportion.

We used VAE to generate single-cell data, described in the main text as follows:

Result: (2) When we trained beta-VAE using real single cell RNA-seq, the Encoder outputs were V (cell type fraction) and W (cell type correlated generative factor), and we added a loss function for minimising the relationship between V and the real cell type fraction. We obtained W for each cell at the end of model training and averaged W for each cell type to represent that cell type. (3) We used the true cell type fraction V calculated by AE with the cell type-associated generating factor W obtained by beta-VAE as input to beta-VAE for generating single-cell data, and deploying unsupervised clustering to denoise and refine the outcomes of the β -VAE

3. *How does the filtering work? Maybe the authors took single cell profiles generated from beta-VAE for clustering analysis, and then removed poorly-classified cells?*

Response: We are thankful for the reviewer's discerning point. During the process of single-cell data generation and clustering with VAE, we encountered the emergence of numerous small cell clusters, each typically comprising 15-20 cells. These clusters, post-dimensionality reduction, were found to be evenly scattered throughout the space. To address this, we employed unsupervised clustering coupled with a size threshold to identify and remove these noisy cells. These noisy cells may have potential biological significance, but they do not help us recover "omitted" cells, so we choose to filter them rather than retain them in subsequent analyses. We show this noisy phenomenon in the extended data Fig. 1e, with additional descriptions in the corresponding places in the text.

It is noteworthy that this approximation introduces a level of noise and bias into the generated data (Extended Data Fig. 1e). Consequently, unsupervised clustering is employed as a data refinement strategy to mitigate the impact of noise and enhance data robustness. We use unsupervised clustering to filter out "noisy" cells, which are identified using community size.

4. *What does "co-CellType" mean in Figure 1B?*

Response: We thank the reviewer for their sharp insight. We regret any ambiguity caused by the term "co-CellType" in Fig. 1b. "Co-CellType" refers to overlapping cell types, which, instead of being depicted as a one-dimensional eigenvector, are now represented by a more apt one-hot matrix to reflect their overlap. To improve transparency and avoid misinterpretation, we have revised the term to "overlap-CellType" and have expanded upon this in the figure legend for a clearer depiction.

Several minor comments:

1. *Typos, such as "similuar" in line 558, should be corrected.*

Response: We thank the reviewer for pointing out the typo in line 558. We have meticulously reviewed the manuscript and corrected this, along with other spelling errors, for the revised version.

2. *Not sure how does reference 22 relate to the formulation of "interrupted" single cells.*

Response: We are grateful for bringing the discrepancy with reference 22 to our attention. Upon revisiting the manuscript, we determined that it was indeed incorrectly cited. We have omitted this reference from the revised manuscript accordingly. Thank you for assisting us in ensuring the accuracy of our citations.

REVIEWERS' COMMENTS

Reviewer #1 (Remarks to the Author):

I want to thank the authors for comprehensively addressing the feedback from the previous round. However, the method section of the manuscript still needs to be improved.

In the main section, the authors cite Shchur et al. (2019) regarding the work on graph neural networks. Still, there also needs to be a citation in the methods section (line 656) because the authors copied sentences from Shchur's paper and replaced individual words. Furthermore, they also copied the formulas but introduced errors:

* In line 666, the authors write that the normalized adjacency matrix is $\hat{A} = \hat{D}^{-1/2} \tilde{A} \hat{D}^{-1/2}$. Shchur et al., however, define this as $\hat{A} = \hat{D}^{-1/2} \tilde{A} \hat{D}^{-1/2}$.

* In line 665, the authors define $\text{GCN}(A, X) = \text{ReLU}(A\hat{A}) \text{ReLU}(A\hat{A}) XW^{(1)} W^{(2)}$. Originally, this was $\text{GCN}(A, X) = \text{ReLU}(\hat{A}) \text{ReLU}(\hat{A}) XW^{(1)} W^{(2)}$.

* The formulas in lines 680 and 667 are incorrect.

* In line 677, the authors refer to "the second term in the third sum". This was originally "the second term in equation (3)".

I recommend that the authors remove the repetitions from Shchur's paper, explicitly cite them in the method section, and only explain what they did differently. The same principle should be applied to all other parts in the method section (i.e., the authors should ensure that they cite prior work sufficiently and do not unnecessarily repeat the content of prior work).

Minor concerns

- A sentence fragment is duplicated
- in line 62-66,
- in line 134-137

- Fig. 1b still refers to "interrupt" cells

Reviewer #3 (Remarks to the Author):

The authors have addressed all my comments.

Reviewer #4 (Remarks to the Author):

The authors answered our questions and revised the manuscript carefully. My concern is that in the Fig 1A there are too many arrows and loops. The workflow is really hard to understand.

Point-to-point replies to the reviewers' comments:

In the following, we present our response to the reviewers' comments. We give *comments (blue)*, *point-by-point answers (green)* to the questions, and in parts copy parts of the text or specific panels (black), which directly correspond to comments or reference to them. All changes in the manuscript text file with **colour highlighting**.

We express our gratitude to the esteemed reviewers for their valuable comments, recommendations, and constructive feedback:

Reviewer#1:

I want to thank the authors for comprehensively addressing the feedback from the previous round. However, the method section of the manuscript still needs to be improved.

Major Concerns

1. *In the main section, the authors cite Shchur et al. (2019) regarding the work on graph neural networks. Still, there also needs to be a citation in the methods section (line 656) because the authors copied sentences from Shchur's paper and replaced individual words. Furthermore, they also copied the formulas but introduced errors:*

** In line 666, the authors write that the normalized adjacency matrix is $\hat{A} = \hat{D}^{-1/2} \hat{A} \hat{D}^{-1/2}$. Shchur et al., however, define this as $\hat{A} = \hat{D}^{-1/2} \tilde{A} \hat{D}^{-1/2}$.*

** In line 665, the authors define $GCN(A, X) = \text{ReLU}(A\hat{A}) \text{ReLU}(A\hat{A}) XW^{(1)} W^{(2)}$. Originally, this was $GCN(A, X) = \text{ReLU}(\hat{A}) \text{ReLU}(\hat{A}) XW^{(1)} W^{(2)}$.*

** The formulas in lines 680 and 667 are incorrect.*

** In line 677, the authors refer to "the second term in the third sum". This was originally "the second term in equation (3)".*

Response: We are grateful for the reviewer's perceptive observations and guidance. Below are the detailed corrections made to address these issues:

Citation Addition:

We have added the citation to Shchur et al. (2019) in the methods section (new line 516) to appropriately acknowledge the source.

Correction of (\hat{A}) Definition:

We have corrected the definition of the normalized adjacency matrix in line 527. The revised text now reads:

$$\begin{aligned} & [\\ \hat{A} &= \hat{D}^{-1/2} \tilde{A} \hat{D}^{-1/2} \\ &] \end{aligned}$$

Correction of GCN Formula:

We have corrected the GCN formula in new line 526. The revised text now reads:

$$\begin{aligned} & [\\ GCN(A, X) &= \text{ReLU}(\hat{A}) \text{ReLU}(\hat{A}) XW^{(1)} W^{(2)} \\ &] \end{aligned}$$

Correction of Formulas in New Lines 541 and 528:

The formula in new line 541 has been corrected to:

$$\left[\mathbb{E} \left\{ \left(u, v \right) \thicksim P_E \left[\log \left(1 - \exp \left(-F_u F_v^T \right) \right) \right] \right\} + \mathbb{E} \left\{ \left(u, v \right) \thicksim P_N \left[F_u F_v^T \right] \right\} \right]$$

The formula in new line 528 has been corrected to:

$$\hat{D}_{ii} = \sum_j \tilde{A}_{ij}$$

Correction of Text Reference:

The content in new line 538 has been modified to "second term in equation (3)" to match the original reference. Additionally, each formula has been labeled in order to improve clarity.

- I recommend that the authors remove the repetitions from Shchur's paper, explicitly cite them in the method section, and only explain what they did differently. The same principle should be applied to all other parts in the method section (i.e., the authors should ensure that they cite prior work sufficiently and do not unnecessarily repeat the content of prior work).*

Response: We have simplified the formulas and remove the repetitions formulas in method section. At the same time, the necessary formulas are still included in our content to clarify the differences between our work and others.

Minor Concerns

- A sentence fragment is duplicated*
 - in line 62-66,*
 - in line 134-137*
 - Fig. 1b still refers to "interrupt" cells*

Response: We thank the reviewer for pointing out the duplicated sentence fragments, all duplication have been deleted, and the "interrupt" in Fig.1b has been revised.

Reviewer #3 (Remarks to the Author):

The authors have addressed all my comments.

Reviewer #4 (Remarks to the Author):

- The authors answered our questions and revised the manuscript carefully. My concern is that in the Fig 1A there are too many arrows and loops. The workflow is really hard to understand.*

Response: We thank the reviewer for their sharp insight. To improve the clarity of Figure 1A and make the workflow of BulkTrajBlend easier to understand, we have made the following revisions:

Color Coding:

We have color-coded the arrows in Figure 1A, using four distinct colors. Each color represents a logical unit of BulkTrajBlend. This differentiation helps to visually separate different parts of the workflow.

Legend Addition:

We have added a legend to the top right corner of Figure 1A. The legend explains what each color represents, providing a clear guide for readers to follow the workflow.